# The WNT target SP5 negatively regulates WNT transcriptional programs in human pluripotent stem cells

Ian J. Huggins[1], Tomas Bos[1], Olivia Gaylord[1], Christina Jessen[1], Brianna Lonquich[1], Angeline Puranen[1], Jenna Richter[1], Charlotte Rossdam[1], David Brafman[2], Terry Gaasterland[3] & Karl Willert[1]

The WNT/β-catenin signaling pathway is a prominent player in many developmental processes, including gastrulation, anterior–posterior axis specification, organ and tissue development, and homeostasis. Here, we use human pluripotent stem cells (hPSCs) to study the dynamics of the transcriptional response to exogenous activation of the WNT pathway. We describe a mechanism involving the WNT target gene *SP5* that leads to termination of the transcriptional program initiated by WNT signaling. Integration of gene expression profiles of wild-type and *SP5* mutant cells with genome-wide SP5 binding events reveals that SP5 acts to diminish expression of genes previously activated by the WNT pathway. Furthermore, we show that activation of SP5 by WNT signaling is most robust in cells with developmental potential, such as stem cells. These findings indicate a mechanism by which the developmental WNT signaling pathway reins in expression of transcriptional programs.

[1] Department of Cellular & Molecular Medicine, University of California San Diego, 9500 Gilman Drive, La Jolla, CA 92093-0695, USA. [2] School of Biological and Health Systems Engineering, Arizona State University, Tempe, AZ 85287, USA. [3] University of California San Diego and Scripps Institution of Oceanography, Scripps Genome Center, 9500 Gilman Drive, La Jolla, CA 92093-0202, USA. Correspondence and requests for materials should be addressed to T.G. (email: gaasterland@ucsd.edu) or to K.W. (email: kwillert@ucsd.edu)

Animal development requires precise coordination among the cells of the embryo to balance cell division and patterning, and thereby ensure the generation of all adult organs and tissues in their proper locations and proportions. Extra-cellular signaling molecules mediate cell–cell communication to control fundamental embryonic processes such as formation of the primitive streak, gastrulation movements, and establishment of the anterior/posterior and dorsal/ventral axes. The WNT/β-catenin signaling pathway (commonly referred to as the canonical WNT pathway), which is highly conserved across all metazoan life forms, is essential for embryonic development and, later in life, for adult tissue homeostasis and regeneration. Deregulation of this pathway causes severe congenital defects, underlies multiple diseases and disorders, and frequently drives oncogenic transformation (reviewed in refs. [1–3]).

Developmental signaling pathways, such as the WNT/β-catenin pathway, initiate signaling cascades that culminate in the expression of many target genes that subsequently mediate developmental programs. To exert temporal control over these highly coordinated developmental processes, these same signaling pathways initiate negative feedback loops that act to desensitize the cell to the signal. Less understood and studied are the mechanisms by which the transcriptional program previously activated by a pathway are diminished and eventually terminated so that a cell can properly respond to subsequent signaling inputs. The prevailing view is that changes in the epigenetic landscape through chromatin modifications and DNA methylation lead to poising and silencing of genes, thereby altering the transcriptional profile of a cell. However, examples of direct connections between developmental signaling pathways and activity of epigenetic modifiers remain scarce.

Recent studies using pluripotent stem cells, such as human embryonic and induced pluripotent stem cells (collectively referred to here as hPSCs), have led to important insights on how developmental programs progress to generate mature cell types, such as cardiomyocytes and pancreatic beta cells (reviewed in ref. [4]). Such studies established that efficient and directed differentiation of hPSCs requires tight temporal control over specific signaling pathways, including those stimulated by WNT, FGF, SHH, NOTCH, and TGFβ. For example, efficient generation of definitive endoderm (DE), a precursor cell population of liver, pancreas, and gut, from hPSCs requires initial activation and subsequent inactivation of WNT/β-catenin signaling[5, 6].

Here we present data supporting a mechanism by which WNT/β-catenin signaling acts to diminish and thereby terminate its own transcriptional program. Using hPSCs, we dissect the temporal changes in gene expression upon WNT pathway activation. The SP5 transcription factor emerged as a critical downstream WNT target that acts to rein in expression of a large swath of genes previously activated by the WNT signal. These findings suggest a mechanism by which a developmental signaling pathway acts to dynamically regulate gene expression.

## Results

**Identification of SP5 as a WNT/β-catenin target gene**. To study the effects of WNT signaling in hPSCs, we analyzed the transcriptomes of cells treated for 12, 24, and 48 h with Wnt3a by high-throughput RNA sequencing (RNA-Seq). Morphological changes consistent with cellular differentiation are apparent by microscopy 48 h post treatment (Fig. 1a). Immunofluorescence (IF) analysis and flow cytometry demonstrate increased expression of differentiation markers, such as SOX17 (Fig. 1b), and a concomitant loss of expression of pluripotency markers, such as FZD7[7, 8] (Supplementary Fig. 1a), SSEA4, and TRA1-81 (Supplementary Fig. 1b). Clustering of significantly differentially

expressed genes (Supplementary Data 1) according to change in percent maximum reads per kilobase per million mapped reads (RPKM) reveal four clear waves of gene expression (Fig. 1c): (i) decreased expression of genes involved in pluripotency and neural differentiation, (ii) transient upregulation of mesendodermal genes, (iii) upregulation of genes involved in primitive endoderm, and (iv) late upregulation genes expressed in DE and the tail bud.

To identify potential downstream transcriptional programs under the control of WNT signaling, we focused on the top 10 upregulated genes encoding known transcription factors (Fig. 1d; Supplementary Fig. 1c). Chromatin immunoprecipitation followed by sequencing (ChIP-Seq) for histone H3 lysine 4 trimethylation (H3K4me3), a mark enriched in promoters of actively transcribed genes[9], reveal SP5 to be potently activated by WNT signaling (Fig. 1d; Supplementary Fig. 1d). Reverse transcription quantitative PCR (qPCR) confirm the induction of SP5 mRNA by recombinant Wnt3a protein in a time- and dose-dependent fashion (Supplementary Fig. 1e, f). AXIN2, a well-known universal target of the WNT pathway[10, 11], is also induced by Wnt3a, albeit at lower levels and only transiently compared to SP5 (Supplementary Fig. 1e, f). Furthermore, expression of SP5, like AXIN2, is induced in a dose-dependent fashion by a small-molecule inhibitor of GSK3β (CHIR98014, Supplementary Fig. 1g), indicating that SP5 expression is likely dependent on transactivation by β-catenin and not another WNT-dependent signaling cascade, consistent with previous studies[12–14].

SP5 encodes a member of the SP/KLF family of Zinc-finger DNA binding proteins that recognizes GC-rich elements, referred to as GC boxes (consensus sequence GGGCGG). SP5, which shares limited domain homology with the ubiquitous SP1 transcription factor (Supplementary Fig. 1h), has previously been described as a WNT/β-catenin target gene[12, 14–18]. Immunoblotting with an antibody directed against the amino-terminus of SP5 (Supplementary Fig. 1h) demonstrates that SP5 protein levels increase substantially upon Wnt3a treatment (Fig. 1e; Supplementary Fig. 1i), consistent with the increase in SP5 mRNA (Supplementary Fig. 1e, f). In contrast, SP1 shows no change in protein levels, again consistent with our RNA-Seq data. IF analysis reveals that SP5 protein is largely localized to the nucleus (Fig. 1f), as expected for a transcription factor.

Studies in mouse embryos and ES cells established that Sp5 and Sp8 act redundantly as key effectors of the Wnt3a/β-catenin pathway[16, 19]. Therefore we mined our RNA-Seq data set for the expression of all known members of the SP/KLF family of transcription factors to determine whether WNT signaling alters the expression of other family members. In contrast to the strong induction of SP5, we observe little or no change in the expression of other SP/KLF family genes, including SP8 (Fig. 1g), suggesting a unique and potentially non-redundant role for SP5 downstream of WNT pathway activation in a human system.

**SP5 regulates differentiation of hPSC**. To study the role of SP5 downstream of WNT signaling in hPSCs, we generated mutations in the SP5 gene using the CRISPR-Cas9 system. The mutagenesis strategy, which was designed to delete the sequence encoding the Zinc-finger domain of SP5 (Fig. 2a), yielded 5 out of 36 clones carrying deletions in both SP5 alleles (Supplementary Fig. 2a, b). We expanded two clones, dZF1 and dZF2 (deleted Zinc Finger) with normal karyotypes (Supplementary Fig. 2c) and confirmed the absence of full-length SP5 proteins by immunoblotting (Fig. 2b; Supplementary Fig. 2d). Levels of a truncated SP5 protein in dZF mutant cells is significantly lower than that of wild-type (WT) SP5 protein, making dominant negative activities unlikely.

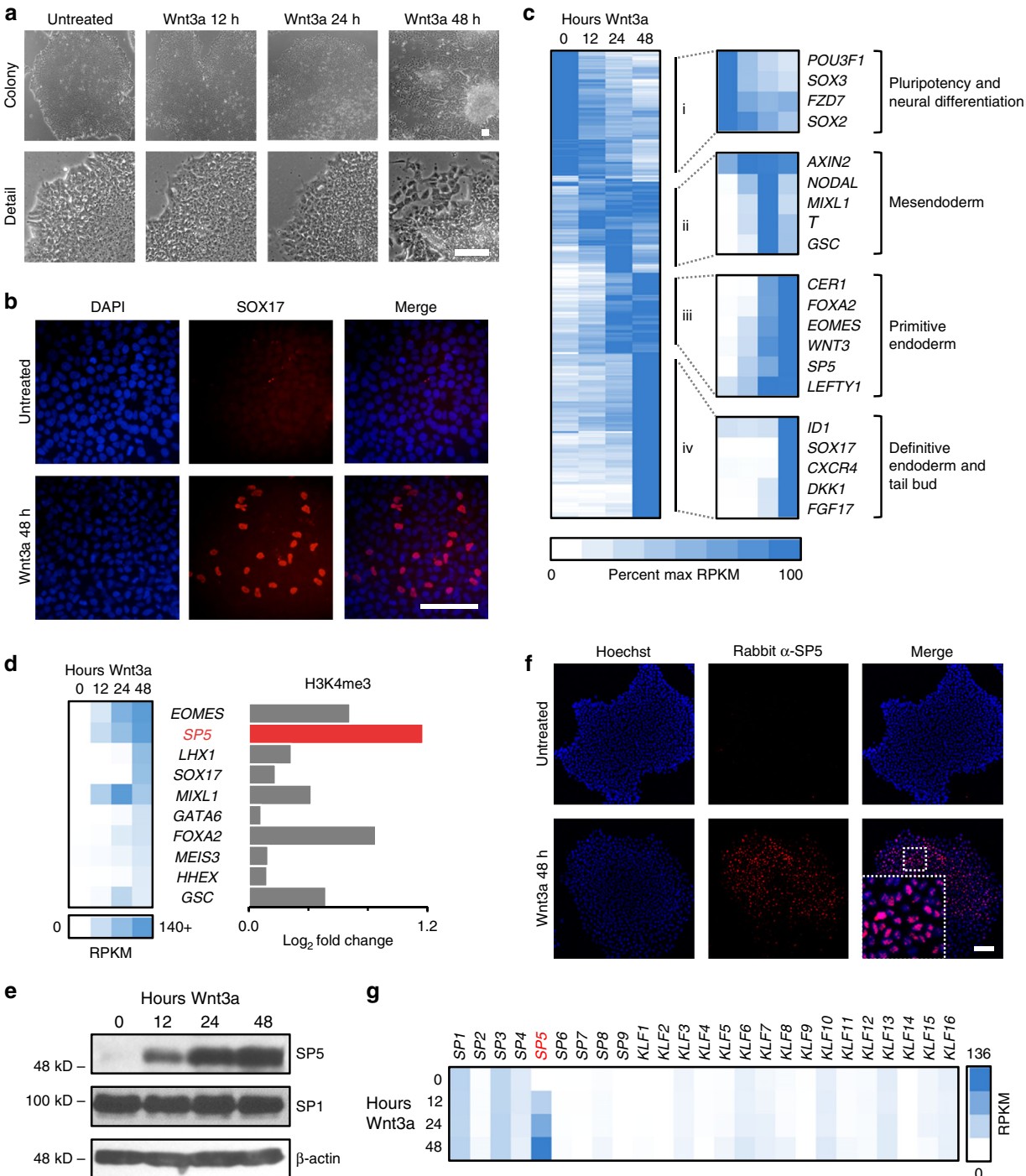

**Fig. 1** Identification of SP5 as a highly responsive WNT/β-catenin target gene in hPSCs. **a** Cell morphology changes in response to Wnt3a treatment. HESCs (H1/WA01) were treated with 1 nM Wnt3a for the indicated times and imaged using phase contrast microscopy. Scale bar = 100 μm. **b** Immunofluorescence (IF) for a differentiation marker. hESCs were treated with 1 nM Wnt3a for 2 days and analyzed by IF for SOX17 expression. Scale bar = 100 μm. **c** Heatmap of genes with significant changes in expression upon Wnt3a treatment. HESCs were treated with 1 nM Wnt3a for the indicated times and RNA was isolated and analyzed by RNA-Seq. About 511 genes with significant differential expression in response to Wnt3a fell into four main clusters i–iv. Refer to Supplementary Data 1 for complete gene set. **d** Transcription factors with greatest differential expression in response to Wnt3a. The top 10 genes encoding transcription factors were sorted by percent maximum reads per kilobase per million mapped reads (RPKM). ChIP-Seq analysis indicated that the SP5 promoter region shows a significant change in the histone mark H3K4me3 upon Wnt3a signaling. Supplementary Fig. 1c, d provides additional information about these transcription factors and display the genome browser tracks of all 10 genes for this histone mark. **e** SP5 protein accumulation in response to Wnt3a treatment. HESCs were treated with 1 nM Wnt3a for the indicated times and nuclear extracts were immunoblotted for SP5, SP1, or β-actin (a loading control). kD kilo Daltons. **f** Immunofluorescence of SP5. HESCs were treated with Wnt3a (1 nM) for 24 h and cells were fixed and stained for SP5 (red, rabbit anti-SP5) and DNA (blue, Hoechst). Untreated cells were incubated with an equivalent volume of WNT storage buffer. Scale bar = 100 μm. **g** Expression of SP/KLF family members in hESCs treated with Wnt3a. RPKM values for all *SP/KLF* genes were extracted from the RNA-Seq data described in **c** and displayed as a heatmap

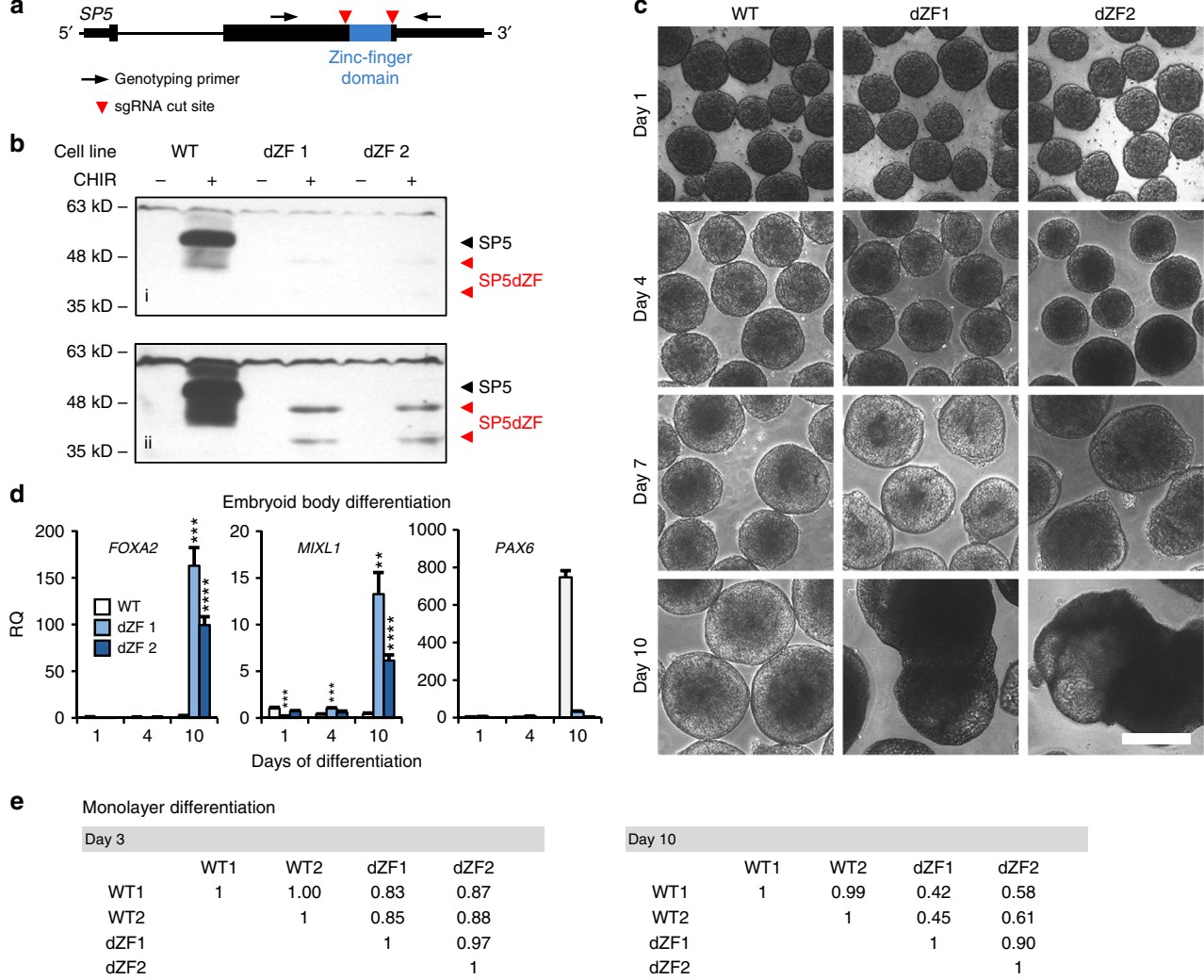

**Fig. 2** SP5 is required for normal hPSC differentiation. **a** Schematic of the *SP5* gene. A deletion of the sequence encoding the C2H2 Zinc-finger DNA binding domain was generated using CRISPR-Cas9 with two guide RNAs (designated by red arrowheads). Gene structure: thin lines for intron and intergenic regions, medium lines for non-coding exonic sequence, and thick lines for open reading frame. The position of genotyping primers is indicated by arrows. **b** Wild-type and mutant SP5 protein detection. Lysates from unstimulated (−) and CHIR-stimulated (+) WT and SP5 mutant (dZF1 and 2) hESCs were analyzed by immunoblotting. Short (i) and long (ii) exposures of the same immunoblot are shown. CHIR = CHIR98014; kD kilo Daltons. **c** Altered EB formation from *SP5* mutant cell lines. SP5 mutant EBs acquire abnormal morphologies at later time points. Scale bar = 100 μm. **d** Altered gene expression in *SP5* mutant EBs. RNA was isolated from EBs at the indicated days after initiation of differentiation and analyzed by qPCR for expression of markers indicative of endoderm (*FOXA2*), mesoderm (*MIXL1*), and ectoderm (*PAX6*). RQ relative quantity. (Error bars are SEM for four technical replicates; Student's t test: **$p < 0.01$; ***$p < 0.001$; ****$p < 0.0001$). **e** Altered gene expression in monolayer differentiation of SP5 mutant cells. WT and SP5 mutant (dZF 1 and 2) hESCs were differentiated through removal of FGF2 and addition of fetal bovine serum. RNA was extracted at 3 and 10 days after initiation of differentiation and analyzed by RNA-Seq. Correlation matrices representing 5253 differentially expressed genes indicate increased divergence of WT (two independent differentiations) and dZF cells at days 3 and 10 of differentiation

As a measure of pluripotency, we injected WT and dZF cells into immune-compromised mice; both cell lines produce teratomas that express derivatives of all three germ layers, as assessed by histological analysis (Supplementary Fig. 2e, f). This experiment demonstrated that dZF cells are pluripotent, however, this assay is not sufficiently quantifiable to detect changes in differentiation potential.

To quantify multi-lineage differentiation potential of *SP5* mutant hPSCs, we generated embryoid bodies (EBs) using established assays. Both WT and dZF hPSCs form tight cell aggregates within 1 day. Upon extended culturing, the dZF EBs are larger in size than their WT counterparts (Fig. 2c; Supplementary Fig. 2g). At 8–10 days of culture, dZF EBs acquire an increasingly rough and uneven morphology compared to the characteristic smooth surface of WT EBs (Fig. 2c; Supplementary Fig. 2h). Gene

expression analysis demonstrates that dZF EBs express significantly higher levels of endodermal markers *FOXA2* and *SOX17*, and of primitive streak markers *MIXL1* and *T* (*BRY*) relative to WT EBs (Fig. 2d; Supplementary Fig. 2i). In contrast, expression of the early ectodermal marker *PAX6* is significantly elevated in WT cells relative to dZF cells (Fig. 2d). Expression of another ectodermal marker, *SOX1*, is only subtly affected (Supplementary Fig. 2i). Together, this gene expression analysis indicates that SP5 is critical for the proper differentiation of hPSCs.

As an additional measure of in vitro differentiation potential, we induced differentiation of adherent WT and dZF cells for up to 10 days. Transcriptome analysis by RNA-Seq reveals that WT and dZF cells diverge increasingly over the course of differentiation (Fig. 2e). These EB formation and monolayer differentiation assays demonstrate that SP5-deficient cells are capable of multi-

lineage differentiation, but exhibit significantly altered lineage allocation compared to WT hPSCs.

**SP5 binds to select locations across the genome**. To identify genes directly regulated by SP5, we performed ChIP-Seq from cells treated with Wnt3a for 24 h, a time point at which SP5 protein was highly elevated (Fig. 1e). Previous work demonstrated that SP5 binds to the same DNA motif as SP1 and competes directly with SP1 for binding to certain loci[17]. Since SP5 is a direct target of WNT signaling and largely resembles a transcriptional

repressor, Fujimura and colleagues hypothesized that WNT downregulates SP1 target genes through the upregulation of SP5, as demonstrated for expression of the *Cdkn1a* (p21) gene. Because of this observed competition between SP1 and SP5, we also performed a SP1 ChIP-Seq analysis on hPSCs treated with Wnt3a. We sequenced co-immunoprecipitated DNA from the SP5 and SP1 pull downs, mapped high-quality reads to the human reference genome assembly GRCH37/hg19 and identified SP5 and SP1 binding sites using the computational method MACs[20].

In contrast to SP1, which binds to a large number of sites across the genome, SP5 binding is significantly more selective

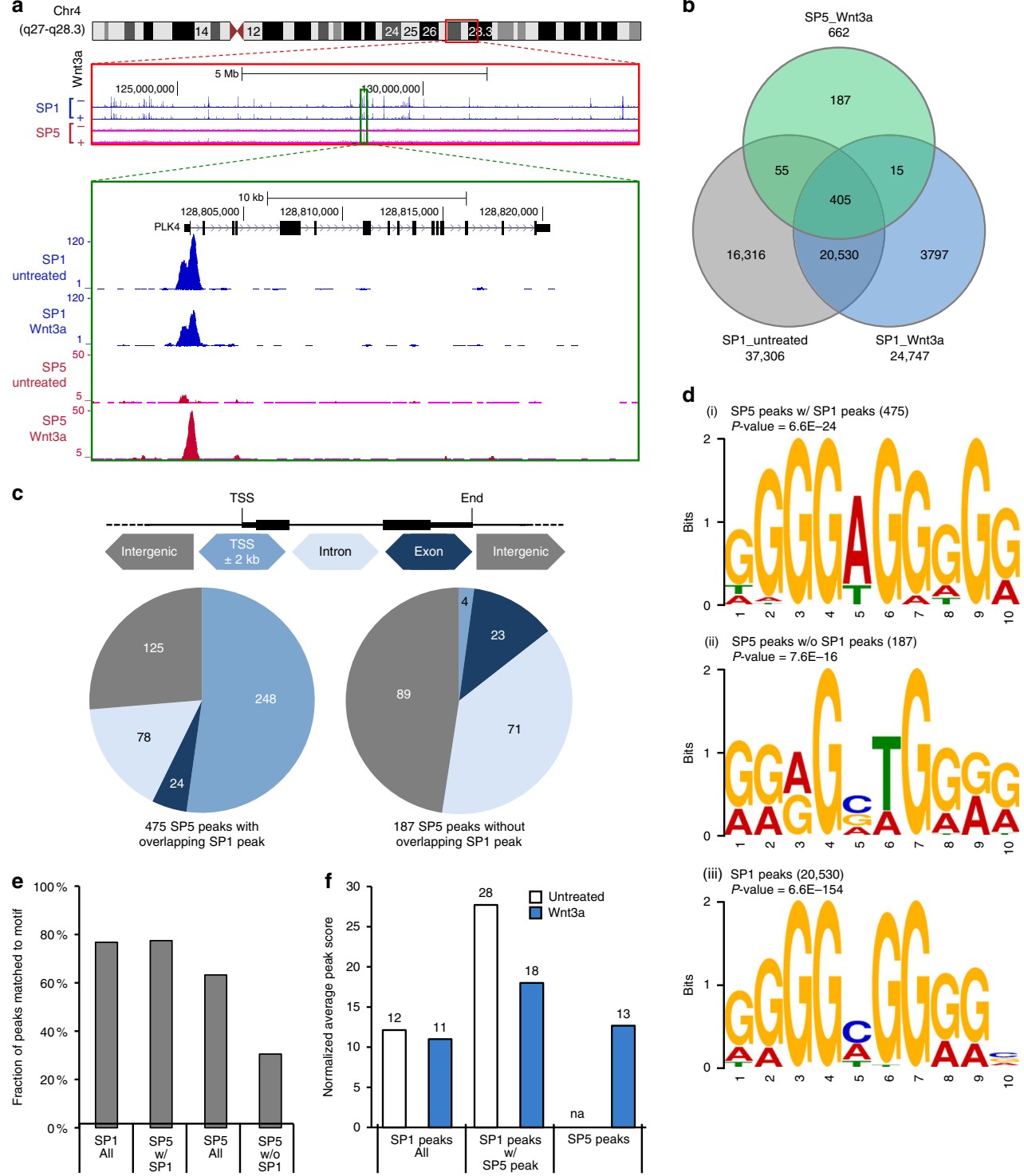

(Fig. 3a). We identify no significant SP5 binding events in untreated cells and 662 SP5 binding events in Wnt3a-treated cells (Fig. 3b; Supplementary Data 2). In contrast, we observe 37,306 and 24,747 SP1 binding events in untreated and Wnt3a-treated cells, respectively (Fig. 3b). These numbers are consistent with available ENCODE data sets for SP1 ChIP-Seq in H1 hESCs[21]. Of the 662 SP5 binding sites, 475 overlap with sites bound by SP1, lending support to the hypothesis that SP5 and SP1 compete for binding at these sites. Of these binding events, a majority map within 2000 bases of transcriptional start sites (TSS, Fig. 3c); in contrast, only 4 of the 187 SP5 binding sites that do not overlap SP1 sites map near TSS (Fig. 3c). Further, SP5 peaks without an overlapping SP1 peak are more distant from TSS than SP5 peaks with an overlapping SP1 peak (Supplementary Fig. 3a).

Motif enrichment analysis of SP5 binding sites overlapping SP1 peaks ($n = 475$) reveals that the most highly represented sequence motif in each peak resembles the previously identified GC-box motif (Fig. 3d, panel i)[22]. The central base at position 5 or 6 of the motif, flanked by Gs, is variable with A more prevalent than C (Fig. 3d, panel i; Supplementary Fig. 3b, panels i and ii). The motif derived from SP5 peaks lacking an overlapping SP1 peak ($n = 187$) differs from this canonical GC box, with a lower frequency of G surrounding positions 5 and 6 (Fig. 3d, panel ii; Supplementary Fig. 3b, panel iii). Summits of SP1 peaks were most highly enriched for the canonical GC box (Fig. 3d, panel iii). As expected, the majority of SP1 and SP5 peaks contains this canonical GC-box motif, whereas a lower fraction of SP5 peaks lacking SP1 peaks contains this motif (Fig. 3e).

We observed that SP1 binding is significantly reduced at sites co-occupied by SP5 upon Wnt3a treatment (Fig. 3f). In contrast, there is no significant decline upon Wnt treatment in SP1 binding at sites that are not occupied by SP5. Therefore, SP5 only competes with SP1 at a select subset of sites and does not act globally to antagonize SP1 binding.

**Integration of gene expression changes with SP5 binding**. To further investigate the function of SP5 downstream of WNT/β-catenin signaling, we compared gene expression profiles in WT and dZF mutant cells treated with Wnt3a. Principal component analysis on this RNA-Seq data confirms that mutant cells are more alike to each other than to WT (Supplementary Fig. 4a). Consistent with results shown in Fig. 2, SP5 mutant cells induce significantly higher levels of expression of genes associated with formation of primitive and DE (Fig. 4a, clusters iii and iv; complete expression data provided in Supplementary Data 3). Genes with significant changes in gene expression upon Wnt3a treatment also exhibit the largest increase in expression in SP5 mutant cells (Fig. 4b), indicating that SP5 exerts an inhibitory effect on these genes.

Consistent with this model, a significant proportion of the genes with highest degrees of expression changes upon Wnt3a stimulation are associated with a SP5 peak. Among the top 2081 genes with significant changes in expression (gene list A, $p$-value $< 0.05$), 161 genes have a nearby SP5 peak, representing a 2.3-fold enrichment relative to all human genes (7.7 % (161 of 2081) vs. 3.3% (774 of 23,135), $p < 1.6E{-}19$). We observe a clear enrichment of genes with SP5 peaks in increasingly refined gene sets, from 10.4% (gene list B, 104 of 1045) to 13.1% (gene list C, 67 of 511) and to 15.5% (gene list D, 9 of 58) (Fig. 4c; complete lists of genes provided in Supplementary Data 4). In addition, gene ontology (GO) analysis (http://geneontology.org) reveals that genes with associated SP5 peaks are enriched for genes of the GO term "primitive streak formation" (Fig. 4d; Supplementary Fig. 4b).

The observation that SP5 peaks are associated with Wnt3a target genes suggest a potential mechanism for the high selectivity of SP5 binding compared to SP1, namely that SP5 interacts with components of the WNT transcriptional response, such as β-catenin. To test this possibility, we employed a proximity ligation method using the engineered peroxidase APEX2[23]. Overexpression of a SP5–APEX2 fusion protein leads to efficient labeling of β-catenin with biotin, indicating that these two proteins are in close proximity (Supplementary Fig. 4c). Serving as a specificity control, β-actin, a highly abundant protein, is not labeled. These observations suggest a mechanism by which SP5 is recruited to genomic binding sites occupied by SP1 in the proximity of β-catenin sites, as is the case for Wnt target genes. These findings are consistent with those of a previous study showing a direct interaction between SP5 and β-catenin[19].

A recent study in mouse cells identified 892 genes with an associated Sp5 peak and differential expression Sp5 overexpression, of which 876 had human homologs measured in our study. Of our list of 746 genes with an associated SP5 peak (Fig. 4c; Supplementary Data 2), 76 genes are common with this mouse gene list (Supplementary Fig. 4d). This overlap contains many of the genes identified to be most significant in our analysis (18 of 67, gene list C), including *AXIN2*, *GATA6*, *NODAL*, *SP5*, *T*, and *TBX3* (Supplementary Data 2). All four lists of genes with expression changes in response to Wnt3a are enriched in SP5 peaks in both the mouse study and ours, indicating the SP5 target genes found in both studies are relevant to regulating response to WNT signaling.

Using qPCR, we confirmed that genes with an associated SP5 peak, such as *SP5* itself (Fig. 4e), *AXIN2*, *AMOTL2*, *GPR37*, *GSC*, *MIXL1*, *NODAL*, and *T* (Supplementary Fig. 5) show significant upregulation in expression upon Wnt3a treatment in SP5 mutant cells. In contrast, genes without an associated SP5 peak, such as *SOX1*, exhibit no significant expression changes between WT and dZF cells (Supplementary Fig. 5). Furthermore, upon prolonged

**Fig. 3** Genome-wide mapping of SP5 binding sites. **a** Representative example of SP1- and SP5-chromatin binding. A segment of chromosome 4 illustrates the highly selective nature of SP5 binding relative to the ubiquitous transcription factor SP1. PLK4 serves a representative example of SP1 and SP5 binding in untreated and Wnt3a-treated cells. **b** A majority of SP5 peaks overlap with SP1 binding sites. A Venn diagram illustrates that 475 SP5 peaks overlapped with SP1 peaks. SP1 bound thousands of targets throughout the genome, whereas SP5 bound a select number of targets, including unique sites and sites shared with SP1. The complete list of SP5 peaks is provided in Supplementary Data 2. **c** A majority of SP5 binding sites map near transcriptional start sites. The schematic of a generic gene structure provides color-coding for the pie charts. Genomic binding locations of SP5 are categorized as transcriptional start site (TSS, defined as 2 kb upstream and downstream of TSS), exon, intron, and intergenic. **d** Motif analysis of SP5 peaks. SP5 binding events were analyzed for overrepresented sequence motifs using MEME. (i) The most overrepresented motif for the 475 SP5 peaks with an overlapping SP1 peak closely matches the GC box. (ii) The most overrepresented motif for the 187 SP5 peaks lacking an overlapping SP1 peak diverges from the canonical GC box. (iii) The most overrepresented motif for SP1 binding summits matches the GC box. **e** Motif enrichment for SP1 and SP5 binding events. The fractions of peaks matching the motif in **d** (i) are provided for all SP1 peaks (SP1 All), SP5 peaks with overlapping SP1 peaks (SP5 w/ SP1), all SP5 peaks (SP5 All), and SP5 peaks lacking an overlapping SP1 peak (SP5 w/o SP1). **f** SP5 and SP1 compete for binding at select sites. The normalized average peak score is provided for all SP1 peaks (SP1 peaks All), SP1 peaks that overlap with SP5 peaks (SP1 peaks with SP5 peaks) and for all SP5 peaks (SP5 peaks) from cells that were untreated or Wnt3a-treated

Wnt3a treatment, expression of Wnt targets, such as *SP5*, declines; this negative feedback is not observed in SP5 mutant cells (Fig. 4f). These data provide evidence that SP5 acts to downregulate expression of genes that are activated upon WNT pathway activation.

**SP5 is a Wnt target in cells with developmental potential**. As observed in various cell types and organisms, activation of WNT/β-catenin signaling leads to the expression of multiple WNT antagonist, including *AXIN2* (often considered a universal WNT target gene), *APCDD1*, *DKK1*, *NKD1*, and *NOTUM*[10, 11, 24–27].

We wished to explore to what extent SP5 expression mirrors the expression of *AXIN2* in a variety of cell types. As expected, the majority of cell lines shows a robust induction of *AXIN2* after Wnt3a stimulation (Fig. 5a; Supplementary Fig. 6a). Cell lines with strong induction of *AXIN2* exhibit poor induction of *SP5*, and vice versa (Fig. 5a). In addition, we observed the highest *SP5* induction in cell populations with stem cell properties, including hPSCs, human neural progenitor cells (hNPCs, derived from hPSCs using established differentiation protocols[28, 29]) and embryonic carcinoma cell lines (NCCIT, PA1). Therefore, in contrast to the universal WNT target *AXIN2*, robust *SP5*

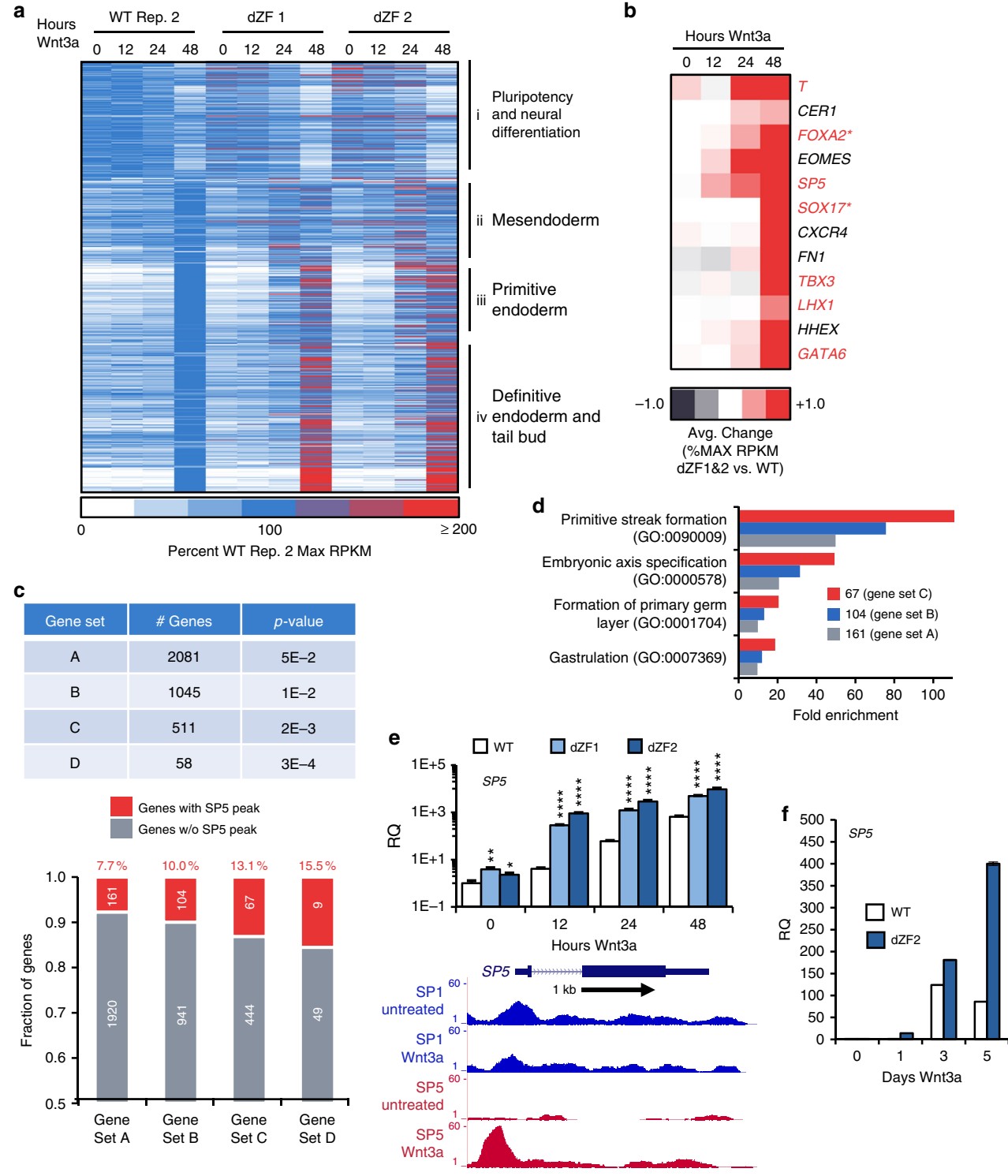

induction correlates strongly with cells that have developmental potential, such as stem cells.

The observed correlation between *SP5* activation and developmental potential suggests that the magnitude of *SP5* induction would decline as hPSCs exit the pluripotent state and acquire an increasingly differentiated phenotype. To test this possibility, we differentiated hPSCs over the course of 28 days and at 1-week intervals evaluated WNT responsiveness by qPCR for *SP5* and *AXIN2* (Fig. 5b). We observed that *SP5* induction by Wnt3a relative to untreated cells is highest in undifferentiated cells, declining sharply in 28-day differentiated cells (Fig. 5c). In contrast, *AXIN2* induction by Wnt3a is only mildly affected (Fig. 5c).

The decline in *SP5* inducibility in differentiated cells may be due to silencing of the *SP5* promoter, loss of a particular TCF/LEF family transcriptional co-activator, or reduced sensitivity to Wnt3a. To address whether this desensitization is due to reduced sensitivity to Wnt3a (e.g., downregulation of Wnt receptors, upregulation of secreted Wnt antagonists (e.g., DKK, SFRP) or of negative regulators of the pathway (e.g., AXIN, NAKED, ZNRF3, RNF43)), we treated cells with the GSK3 inhibitor CHIR98014 (CHIR) to activate the pathway downstream of Wnts and their receptors. Although CHIR, like Wnt3a, potently activates *SP5* expression in undifferentiated cells, it fails to do so in 28-day differentiated cells (Fig. 5d), demonstrating that the decline in *SP5* induction is by some other means than reduced sensitivity to Wnt3a. As expected for hPSC differentiation, expression of *POU5F1* and *NANOG*, key regulators of the pluripotent stem cell state[30–32], declines significantly over the course of the 28-day differentiation (Fig. 5c). Furthermore, we observe dynamic changes in expression of germ layer markers *SOX17* (endoderm), *T* (mesoderm) and *PAX6* (ectoderm), indicating loss of pluripotency (Supplementary Fig. 6b). In addition, we observe a similar suppression in *SP5* (but not of *AXIN2*) inducibility in hPSC-derived neurons relative to their parental hNPCs (Supplementary Fig. 6c), further supporting the hypothesis that *SP5* is a WNT/β-catenin target gene in cells with developmental potential.

To identify the mechanisms by which SP5 induction is silenced over the course of differentiation, we examined the upstream regulatory regions of the *SP5* gene in undifferentiated and differentiated cells. Using chromatin immunoprecipitation followed by PCR (ChIP-PCR), we examined levels of histone H3 lysine 4 di-and tri-methylation (H3K4me2 and H3K4me3), which mark poised[33] and actively transcribed chromatin[9], as well as levels of histone H3 lysine 27 di- and tri-methylation (H3K27me2 and H3K27me3), which mark facultative heterochromatin[34], in the *SP5* promoter region. This analysis reveals a significant reduction in H3K4me2 and H3K4me3 at two loci near the transcriptional start site of *SP5* following differentiation (Fig. 5e), and subtle to insignificant changes in the repressive marks

H3K27me2 and H3K27me3 (Supplementary Fig. 6d, e). To test the possibility that *SP5* is silenced through the methylation of its promoter, we measured the change in CpG methylation in the *SP5* promoter (Supplementary Fig. 6f) at the beginning and end of differentiation. We observe no significant differences in *SP5* promoter methylation levels between undifferentiated and differentiated cells (Supplementary Fig. 6g). Therefore, neither repressive chromatin marks nor promoter methylation explain reduced *SP5* induction in differentiated cells. Rather, loss of the critical histone marks H3K4me2 and H3K4me3 likely accounts for the loss of *SP5* induction in response to Wnt3a.

## Discussion

Termination of the transcriptional program induced by developmental signaling pathways is critical to ensure tightly orchestrated developmental events. While WNT/β-catenin signaling has known mechanisms to antagonize components of the pathway, including WNT receptors and the WNT protein itself, and thereby desensitize cells to subsequent WNT signals, little was known about how key target genes were subsequently downregulated. Here, we have shown that the WNT target gene *SP5* has a key role and acts to downregulate expression of many WNT target genes, including *SP5* itself, thereby dampening the transcriptional response initiated by WNT signaling. Such dampening of gene expression is critical to ensure progression and completion of developmental programs. Our findings indicate a mechanism by which developmental signaling pathways are regulated (summarized in Fig. 6). Prior studies have focused on the mechanisms by which developmental programs are initiated and have established that the ability to interpret appropriately inductive signals from the environment, a phenomenon referred to as developmental competence, is determined by poising the transcriptional state of critical developmental regulators.

A common theme in developmental signaling is the activation of negative feedback systems that ensure the developmental program is under tight temporal control. In the case of the WNT/β-catenin signaling pathway, this negative feedback system involves mechanisms that act at multiple levels on the components of the pathway, including the signaling molecule WNT (SFRPs, DKK, NOTUM), the FZD receptors (RNF43, ZNRF3, APCDD1), and the intracellular signaling cascade (AXIN2, NKD). While this system ensures signal desensitization, it does not act on the expression of many of the previously activated target genes. SP5 provides a previously unrecognized mechanism by which a cell downregulates and thereby fine-tunes the expression of complex transcriptional programs.

A recent study raised the question whether SP5 acts globally to antagonize SP1 target gene expression[17]. Our findings demonstrated that SP5 does not compete with SP1 for target site binding

Fig. 4 SP5 regulates expression of developmental regulators. **a** Altered gene expression in WNT-treated SP5 mutant cells. A heatmap of gene expression illustrates differences between WT and SP5 mutant (dZF1 and 2) hESCs. Genes are in order of the clusters identified in Fig. 1c; expression is given as percent WT maximum RPKM (0, white; 100, blue; ≥100, red). Refer to Supplementary Data 3 for complete gene list. **b** Representative upregulated genes. The average change in percent maximum RPKM of dZF 1 and dZF 2 vs. wild-type illustrates upregulation of many developmental regulators (−1, indigo; 0, white; 1.0, red). Genes in red contain a SP5 peak. *SP5 peaks near *FOXA2* and *SOX17* were called at a lower confidence ($p < 10^{-4}$ vs. $p < 10^{-5}$) and were not included in the list of 662 SP5 peaks. **c** SP5 peaks are enriched among genes with the most significant fold changes upon WNT treatment. For each gene set (A through D), the table provides the p-value for the least significantly changed gene (as described by Trapnell et al.[52, 53]). Refer to Supplementary Data 4 for gene lists. The graph indicates the fraction of genes with SP5 peaks among gene sets A through D. **d** Genes with SP5 peaks are associated with early embryonic development. Gene ontology (GO) analysis reveals that genes with SP5 peaks of gene set C are highly enriched for genes associated with "primitive streak formation". **e** Validation of upregulated genes. WT and dZF mutant hESCs were treated with 1 nM Wnt3a for the indicated times and RNA was isolated and analyzed by qPCR for expression of *SP5*. Genome browser tracks illustrate the increase of SP5 binding near the transcriptional start site. Additional examples are provided in Supplementary Fig. 5. Arrows indicates direction of transcription. RQ relative quantity. (Error bars are SEM for four technical replicates; Student's t test: *$p < 0.05$; **$p < 0.01$; ****$p < 0.0001$. **f** Extended Wnt3a treatment of hPSC. WT and SP5 mutant (dZF2) cells were treated with Wnt3a for the indicated times and RNA was analyzed by qPCR for *SP5* expression

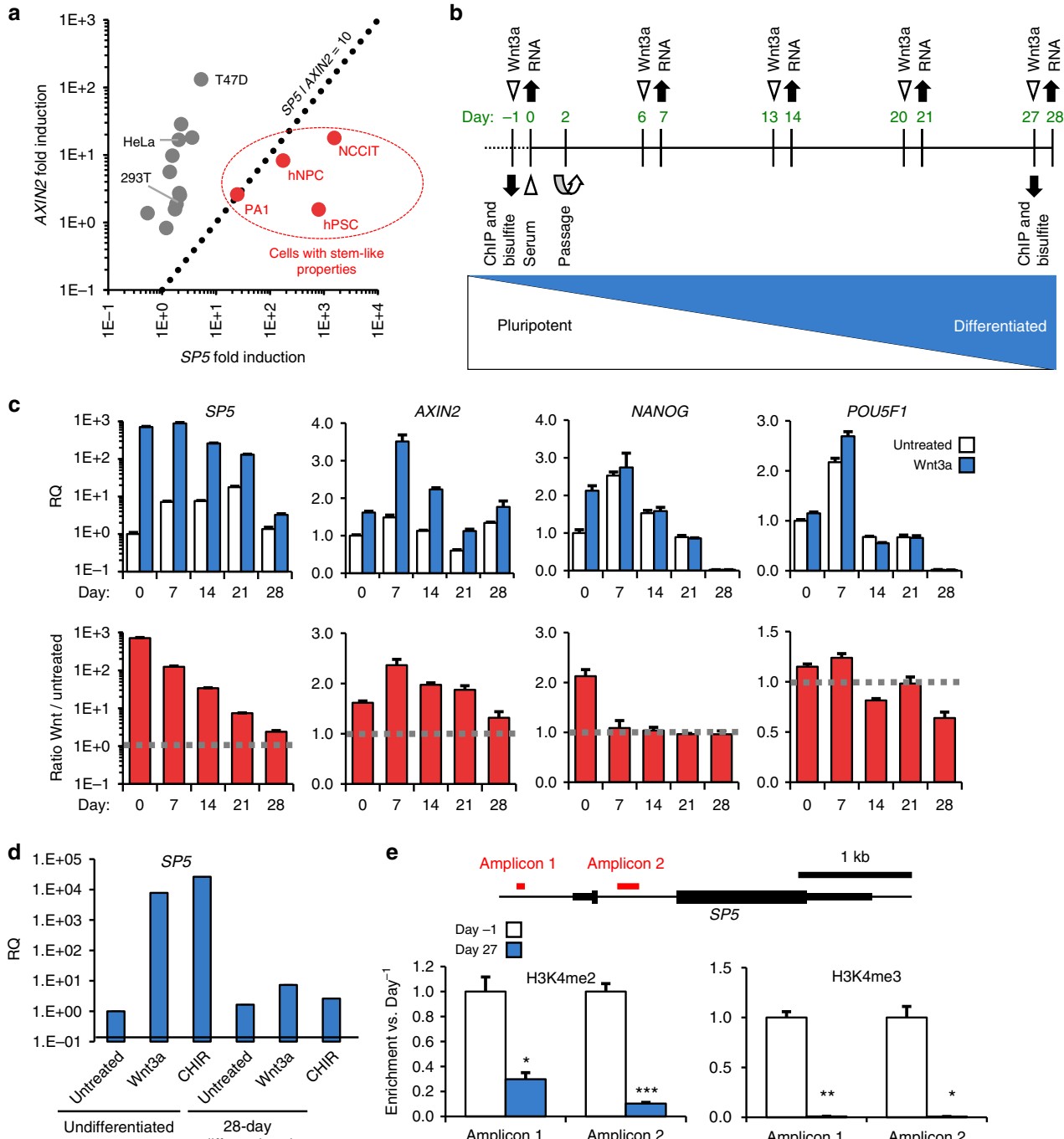

**Fig. 5** Robust SP5 induction is associated with developmental potential. **a** *SP5* and *AXIN2* induction in multiple cell lines. Multiple cell lines (Supplementary Fig. 6a) were treated with Wnt3a (1 nM) for 24 h and expression of *AXIN2* and *SP5* was quantified by qPCR. Red dots indicate cell types with significant *SP5* induction relative to *AXIN2* induction (dashed line demarcates boundary, where *SP5* induction exceeds *AXIN2* induction by 10-fold). **b** Schematic of experimental design. Undifferentiated hESCs (H1/WA01) were differentiated for 28 days, cells were treated with Wnt3a on the indicated days, and RNA was isolated 1 day later for analysis by qPCR. Chromatin and genomic DNA were isolated from undifferentiated and 27-day differentiated cells, and analyzed by ChIP and bisulfite sequencing. **c** WNT responsiveness over a differentiation course. Cells were treated with Wnt3a on the indicated days of differentiation and RNA was analyzed by qPCR for levels of *SP5*, *AXIN2*, *NANOG*, and *POU5F1* mRNAs. RQ relative quantity. The bottom graphs represent the ratios of relative gene expression in Wnt3a-treated vs. untreated cells. (Error bars are SEM for four technical replicates. **d** Activation of WNT signaling by GSK3 inhibition fails to overcome desensitization of SP5 responsiveness in differentiated cells. RNA from undifferentiated and 28-day differentiated cells treated with either buffer (untreated), Wnt3a, or CHIR98014 (CHIR) was analyzed by qPCR for SP5 expression. **e** ChIP-PCR analysis of the *SP5* promoter region. Cross-linked chromatin was isolated from cells prior to differentiation (Day-1) and on Day 27 of differentiation and immunoprecipitated with antibodies to the activating histone marks H3K4me2 (left panel) and H3K4me3 (right panel). Enrichment of immunoprecipitated chromatin was quantified by PCR of two regions of the SP5 promoter, amplicons 1 and 2, the positions of which are indicated in the *SP5* gene schematic. Promoter marks H3K3me2 and H3K3me3 are greatly depleted in differentiated cells relative to undifferentiated cells. (Error bars are SEM for four technical replicates; Student's *t* test: *p < 0.05; **p < 0.01; ***p < 0.001)

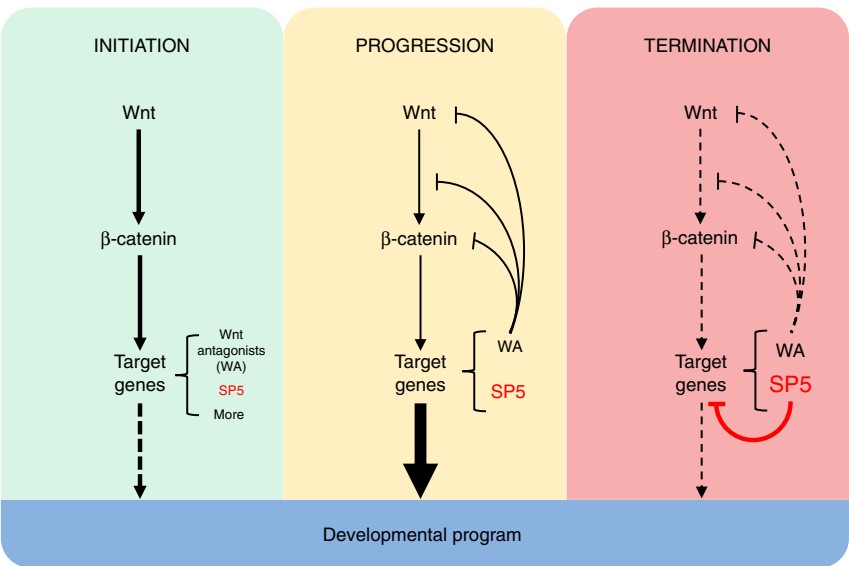

**Fig. 6** Model for SP5's role in terminating a developmental program. WNT signaling is initiated when WNT engages its receptor encoded by FZD, leading to stabilization of β-catenin and subsequent activation of target gene expression. WNT target genes encode WNT antagonists (WA), which serve to downregulate the WNT signaling input as other target genes mediate the WNT effect on a developmental program. During this stage of signaling, SP5 protein accumulates and then competes with SP1 for binding at many WNT target gene sites. The net effect is that SP5 acts to diminish and terminate the transcriptional program previously activated by WNT

on a global scale in hPSCs, but rather is concentrated on a select subset of genes, a significant fraction of which were differentially expressed upon WNT pathway activation. Therefore, SP5 primarily acts on dynamically expressed genes and does not disrupt expression of ubiquitous housekeeping genes that are under control of SP1.

The role of Sp5 downstream of Wnt signaling was recently examined in mouse development and embryonic stem cells[19]. That study concluded that Sp5 acts in a feed-forward loop to robustly activate select Wnt target genes, a conclusion that is in stark contrast to our hypothesis that SP5 acts to rein in expression of WNT target genes. Aside from the obvious differences in systems (embryo vs. embryonic stem cells) and organisms (mouse vs. human), we offer several possible explanations to address the disparate findings. First, RNA-Seq and ChIP-Seq experiments in the Kennedy et al.[19] study involved overexpression of Sp5 protein, which may produce activities normally not associated with endogenously expressed protein. Our studies were performed using endogenously expressed SP5, the expression of which was dynamically regulated by activation of the WNT pathway. This approach allowed us to observe the dynamic changes in gene expression downstream of WNT signaling such that the antagonistic activity of SP5 is only observed later during the signaling program.

Second, in the Kennedy et al.[19] study, loss-of-function phenotypes were evaluated in the context of Sp5 and Sp8 double knockout (DKO). Such DKO mESCs exhibited significantly lower induction of the Wnt target gene *T* upon Wnt pathway activation, in contrast to our *SP5* knockout hESCs where WNT target genes, including *T*, were consistently upregulated upon Wnt pathway activation. Although Sp5 and Sp8 are highly related to each other and may act redundantly during mouse development, these two proteins may carry distinct functions. Interestingly, we did not observe robust transcriptional activation of *SP8* as previously observed upon Wnt pathway activation in mESCs[16], suggesting that in hPSCs SP5 function is not redundant with SP8.

Finally, our observations further highlight the clear distinctions between mESCs and hPSCs that have been the subject of many studies and reviews (reviewed in ref. [35]). It is noteworthy that

naive mESC exhibit higher basal WNT signaling activity relative to the primed hPSC used in these studies[29, 36, 37]. Taken together, the variable results described by the two studies can be attributed in part to differences in systems and approaches. Nonetheless, it should be emphasized that both studies support SP5 as a modulator of response to WNT signaling. Importantly, intersecting mouse and human gene sets revealed that highly similar transcriptional programs are under WNT-SP5 control.

This function of SP5 is potentially relevant to other cell populations capable of self-renewal and differentiation. We found that induction of SP5 by WNT/β-catenin signaling was most prominent in cell types with stemness properties, including hPSCs, hPSC-derived neural progenitor cells (hNPCs), and pluripotent embryonal carcinoma cell lines (PA1 and NCCIT). The levels of induction were sharply reduced upon differentiation of hPSCs to a fibroblast-like population and of NPCs to neurons. Likewise, all tested cell lines that lack stemness properties exhibited a relatively minor to absent induction of *SP5* upon WNT treatment. While the mechanism(s) underlying loss of *SP5* induction are not completely understood, we showed that a histone mark, H3K4me2, which marks poised[33] and actively transcribed chromatin[9], is lost as hPSCs differentiate. The observed correlation between *SP5* inducibility and stemness contrasts with *AXIN2*, often considered a universal WNT/β-catenin target with little variability in induction across cell lines. Our data establish a strong correlation between pluripotency—or developmental potential—and *SP5* inducibility. It is also noteworthy that *Sp5* was identified as a top WNT target gene in Lgr5[+] intestinal stem cells (ISCs)[15], which receives a WNT signal from adjoining paneth cells[38], raising the possibility that *SP5* inducibility is not restricted to pluripotent stem cell populations but also occurs in multipotent stem cells, such as ISCs. It will be interesting and important to further examine *SP5*'s function as a stem cell-specific WNT target gene using genetic lineage tracing experiments, as has been performed for *Axin2*[39–43]. Furthermore, it will be important to examine to what extent this WNT-SP5 regulatory system is altered in pathological states, such as cancer, where SP5 expression is commonly upregulated[44, 45], a situation that likely impacts the behavior of stem cells.

## Methods

**Cell lines and culture conditions.** Human pluripotent stem cell lines H1 (WA01/NIH Registration Number 0043) and H9 (WA09/NIH Registration Number 0062) were cultured in E8 culture medium[46] on Matrigel (BD Biosciences). Upon passage, cells were dissociated with Accutase (Innovative Cell Technologies) to single cells and seeded at 2000 cells cm$^{-2}$ in the presence of the Rock inhibitor Y-27632, 5 μM (Enzo Life Sciences). Cells were fed daily. Medium was changed 50% daily. For the formation of EBs, cells were dissociated to single cells and seeded in ultra-low-attachment culture dishes at 375,000 cells mL$^{-1}$ with 5 μM Y-27632 in E6 medium (E8 without TGF-β1 and FGF2). Fresh E6 medium without Rock inhibitor was added daily. Differentiation of hPSCs in monolayer cultures was achieved by culturing cells in E6 medium supplemented with 20% fetal bovine serum (Omega Scientific) and Penicillin–Streptomycin (Thermo Fisher Scientific). Wnt3a protein used in these studies was purified by four-step column chromatography from conditioned medium harvested from cells engineered to overexpress Wnt3a[47, 48]. Untreated cells received an equal volume of WNT storage buffer (PBS, 1% CHAPS, 1 M NaCl). Human neural stem cells (HUES7/NIH Registration Number 0020) were a gift from Dr. Martin Marsala and derived by dual inhibition of SMAD signaling[28]: EBs were formed over 5 days in the presence of 50 ng mL$^{-1}$ recombinant mouse noggin (R&D Systems) and 0.5 μM Dorsomorphin (Tocris Bioscience). After 7 days, neural rosettes were isolated, dissociated into single cells, and plated onto poly-L-ornithine (10 μg mL$^{-1}$) and mouse laminin (5 μg mL$^{-1}$)-coated dishes with 10 ng mL$^{-1}$ mouse FGF2 and 10 ng mL$^{-1}$ mouse EGF2. Resulting cells were cultured in DMEM/F12 medium (Gibco) with Glutamax (Life Technologies), N2 supplement (Life Technologies), B-27 supplement (Life Technologies), and recombinant FGF2 (20 ng mL$^{-1}$) (Peprotech). Mesodermal progenitor cells were derived and cultured in the Willert laboratory[49]. All other cell lines (BT-549, ATCC-HTB-122; H1299, ATCC-CRL-5803; HEK-293, ATCC-CRL-1573; HeLa, ATCC-CCL-2; K-562, ATCC-CCL-243; NCCIT, ATCC-CRL-2073; OVCAR-3, ATCC-HTB-161; OVCAR-4, RRID:CVCL_1627; PA1, ATCC-CRL-1572; SK-OV-3, ATCC-HTB-77; SW480, ATCC-CCL-228; T-47D, ATCC-HTB-133) were cultured as specified by ATCC (http://www.atcc.org). All cell lines were tested for mycoplasma contamination. For proximity ligation experiments, HEK-293 cells were transfected with SP-APEX2, and 48 h post-transfection cells were incubated for 30 min with Biotin-phenol (Adipogen Life Sciences) to a final concentration of 500 μM. The labeling reaction was initiated by adding 1 mM H$_2$O$_2$ and incubated for 1 min, after which cells were lysed as described below.

**Plasmids.** The following plasmids were used in these studies: pCMMP IRES-GFP (modified pCMMP-EnvA/RG-IRES-GFP from Callaway Lab, Salk Institute for Biological Studies); pCMMP SP5-IRES-GFP (modified pCMMP-EnvA/RG-IRES-GFP from Callaway Lab, Salk Institute for Biological Studies); pGIPZ shSP5-GFP #3 (GE Dharmacon, Clone ID: V3LHS_372625); pGIPZ shControl-GFP (GE Dharmacon, Catalog #: RHS4346); psPAX2 (Addgene); pVSVG (Addgene); pCMVGP (Addgene); pCas9-EGFP (Cowan Lab, Harvard University); pGRNA SP5dZF 3′ (modified pGRNA from Cowan Lab); pGRNA SP5dZF 5′ (modified pGRNA from Cowan Lab); pLEX-SP5-APEX2 (the SP5 open reading frame flanked by Gateway attB sites was synthesized as a gBlock (Integrated DNA Technologies), recombined into pDONR221 and subsequently transferred into a modified pLEX307 plasmid containing the ORF for APEX2 such the transgene encodes SP5 protein with a C-terminal APEX2).

**DNA methylation analysis.** Genomic DNA was prepared with the NucleoSpin Tissue kit (Clontech) according to manufacturer recommendations and bisulfite converted using the EZ DNA Methylation Kit (Zymo Reseach) according to manufacturer recommendations. Regions to be sequenced were amplified by PCR, cloned using the Zero Blunt TOPO PCR Cloning Kit (Thermo Fisher Scientific) and multiple clones for each region were sequenced. Methylation analysis was performed using BiQ software. Primer sequences are provided in Supplementary Table 1.

**CRISPR-Cas9 hPS cell targeted mutagenesis.** Deletions in the SP5 gene were generated using CRISPR-Cas9 technology[50]. Single-guide RNA targets flanking the zinc-finger coding region of the SP5 locus were selected using CHOPCHOP (https://chopchop.rc.fas.harvard.edu/about.php) and cloned into pGRNA (see plasmid list below). Cells were transfected with GeneIn Reagent (GlobalStem) according to manufacturer recommendations. Briefly, two wells of a six-well plate were seeded 24 h pre-transfection with 237,000 cells per well in E8 + Y-27632 (5 μM). On the day of transfection, pCas9-GFP (2 μg), pGRNA SP5dZF 5′ (1 μg, 5′-GCAGACCAAGGCCCCCCUGGCGG-3′) and pGRNA SP5dZF 3′ (1 μg, 5′-GGUUAAGCGGGAGGACGCGCGGG-3′) were combined in OptiMEM medium (400 μL) with GeneIn Red (8 μL) and Blue (8 μL) reagents. Cells were washed once with dPBS and given fresh E8 culture medium with fetal bovine serum (10%) and no penicillin–streptomycin. Transfection mix (200 μL) was added dropwise to the cells, which were returned to a humidified 37 °C, 5% CO$_2$ incubator for 12 h. Cells were washed once with dPBS and fed fresh E8 without penicillin–streptomycin. GFP-positive cells were isolated by FACS and seeded on a Matrigel-coated dish in E8 medium at a density that allowed for manual isolation

of single-cell colonies after ~10 days of standard hPSC culture. Single-cell clones were genotyped by PCR, and chromosome numbers were determined by counting of metaphase spreads (Cell Line Genetics, Madison, WI). PCR Genotyping Primers: Forward: 5′-ACAAAGAGGCCTGGTGTTGG-3′, Reverse: 5′-CATTTTGGGAGGCAGGCAAC-3′. The expected size of amplicons for WT sequence was 609 bp, and for the zinc-finger deleted sequence was ~216 bp, based on predicted cut site locations.

**Embryoid body morphological analyses.** For size quantification, EBs cultured in non-adherent tissue culture dishes were swirled gently, allowed to settle and imaged in the center of the plate on an inverted phase contrast light microscope daily for 5 days. The largest diameters of visible EBs were measured using ImageJ software. For histological analyses, Day 10 EBs were fixed in paraformaldehyde, embedded in paraffin, sectioned, stained with hematoxylin and eosin and imaged on a light microscope.

**Teratoma assay.** Pluripotent stem cells were washed with PBS, dissociated with Accutase (Life Technologies), centrifuged at 100×g for 3 min. Cell pellet was resuspended in 300 μL Matrigel (BD Biosciences) to final volume of 600 μL. Cells were kept on ice prior to injection. Cells were injected subcutaneously at two locations per nude mouse: the shoulder and hind leg. Injections were performed using a 1 mL syringe with a 28-gauge needle. Teratoma formation was monitored over a period of 4–8 weeks until tumors grow to an approximate size of at least 10 mm. Animals were killed and tumors were dissected and fixed in 4% paraformaldehyde at 4 °C. After 48 h, teratomas were embedded in paraffin before sectioning. Five-micron sections were stained with hematoxylin and eosin according to standard protocols. Animal protocol was previously approved by the University of California San Diego Institutional Animal Care and Use Committee (Protocol Number S09005, PI A. Muotri, UCSD). Random images were taken from each sample before analysis. Germ layer distribution was semi-quantified by visual inspection of H&E-stained sections in a blinded fashion. Differentiated tissue with defined areas in the teratoma were classified into ectoderm, endoderm, and mesoderm. The remaining areas remained unclassified.

**Preparation of retrovirus and lentivirus.** Retrovirus was prepared by transfection of HEK293T cells with the appropriate viral plasmids along with pCMVGP and pVSVG. Lentivirus was prepared by transfection of HEK293T cells with the appropriate viral plasmids along with psPAX2 and pVSVG. Supernatants were collected and filtered through 0.45 μm filter and virus was concentrated by ultra-centrifugation in an Optima L-80 XP Ultracentrifuge (Beckton-Dickinson) for 2 h at 70,000×g in a SW-40 Ti rotor. Pellets were resuspended overnight at 4 °C in DMEM (Gibco) and frozen at −80 °C until transduction.

**RNA expression analysis.** RNA expression was measured by quantitative real-time PCR (qPCR) on a CFX384 thermocycler (Bio-Rad). RNA was collected by TRIzol Reagent (Life Technologies) or by column purification with RNeasy PLUS (Qiagen) or NucleoSpin RNA II (CloneTech) Kits according to manufacturer's recommendation. Total RNA (50 ng μL$^{-1}$ final concentration) was used to generate first strand complimentary DNA (cDNA) using qScript cDNA Supermix (Quanta). For amplification, cDNA (0.5 μL per well), primers (0.4 μM final concentration), and SensiFAST Hi-ROX qPCR Master Mix (Bioline) were mixed and dispensed 5 μL per replicate, 4 replicates in a 384-well qPCR plate. Thermal cycler parameters: initial denaturation, 95 °C for 3 min followed by 40 cycles of annealing and extension, 60 °C for 5 s and denaturation, 95 °C for 15 s. Primers were validated by melt curve analysis (60–95 °C, 5 min) and gel electrophoresis of products. Data were analyzed and statistical analyses performed using CFX Manager (Bio-Rad) and plotted with Microsoft Excel. All gene expression was normalized to the expression of an appropriate internal control gene (18 S, GAPDH, RPL13A, RPL37A, and TBP). QPCR primer sequences are provided in Supplementary Table 1.

**Immunoblotting.** For immunoblot analysis of protein expression, cells were washed once with dPBS, dissociated with PBS + 5 mM EDTA and pelleted at 1000 G, 4 °C for 2 min. Cell pellets were resuspended in 500 μL hypotonic lysis buffer (20 mM Tris-Cl, pH 7.4, 10 mM NaCl, 3 mM MgCl$_2$) and incubated on ice for 15 min. About 25 μL 10% IGEPAL CA-630 (Sigma-Aldrich) was added and tubes were vortexed for 10 s. Samples were centrifuged at 1000×g for 10 min at 4 °C. The supernatant (=cytoplasmic fraction) was removed and the pellet (=nuclear fraction) was resuspended in 50 μL cell extraction buffer (1% Triton X-100, 0.1% SDS, 0.5% deoxycholate, 100 mM NaCl, 1 mM EDTA, 1 mM EGTA, 10% glycerol, 1 mM NaF, 20 mM Na$_4$P$_2$O$_7$, 2 mM Na$_3$VO$_4$, 10 mM Tris-Cl, pH 7.4) freshly supplemented with 1 mM Phenylmethylsulfonyl Fluoride (PMSF, Sigma-Aldrich P7626) and protease inhibitor cocktail (Sigma-Aldrich, P8340) and incubated on ice for 30 min, vortexing every 10 min. Samples were centrifuged at 20,000×g, 30 min at 4 °C. The supernatant representing the nuclear fraction, was transferred to a fresh tube on ice and protein concentration was quantified by Coomassie protein assay (Thermo Fisher Scientific). About 20 μg of total protein per lane was resolved by sodium dodecyl sulfate–polyacrylamide gel electrophoresis (SDS–PAGE), transferred to nitrocellulose, blocked for 1 h at room temperature in

blocking buffer (TBS, 0.2% Tween-20, 1% BSA, 3% non-fat dry milk) and probed with the appropriate primary antibodies (see below) in blocking buffer overnight at 4 °C. Protein was detected following incubation with secondary antibodies for 1 h at room temperature, incubation with Western Lighting ECL Reagent (Perkin-Elmer), and exposure to autoradiography film. For proximity ligation experiments with SP-APEX2, transfected HEK-293 treated with Biotin-phenol and H2O2 (see above under "Cell Lines and Culture Conditions" section) were washed twice with Quencher Solution (10 mM Sodium Ascorbate, 5 mM Trolox, and 10 mM Sodium Azide in 1× PBS) and twice with PBS. Cells were collected and lysed in RIPA buffer containing 10 mM Sodium Ascorbate, 5 mM Trolox, and 10 mM Sodium Azide for 10 min on ice. Lysates were centrifuged for 10 min at 12,000×g, 4 °C. About 20 µL of cleared supernatant was stored at −80 °C and used as input. To the remaining supernatant, 200 µL Streptavidin beads (New England Biolabs) were added and incubated for 1 h with rotation. Beads were washed twice with RIPA buffer, once with 2 M Urea in 10 mM Tris-HCl pH 8, twice with RIPA buffer and twice with PBS. Proteins were eluted from the beads in 50 µL elution buffer (200 mM NaCl, 50 mM Tris-HCl pH 8, 2% SDS, and 1 mM Biotin in water) for 30 min at 60 °C[51]. Protein samples were resolved by SDS–PAGE, transferred to nitrocellulose and probed with the indicated antibodies.

**Immunofluorescence.** Cells were fixed with BD Cytofix (BD Bioscience) for 10 min at 22 °C, washed twice with dPBS (Cellgro) and permeabilized in dPBS + 0.2% Triton X-100 for 1 h at 22 °C followed by two additional dPBS washes. For antigen recovery, required for SP5 staining, cells were incubated with dPBS + 1% Sodium Dodecyl Sulfate for 10 min at 22 °C and washed three times with dPBS. Cells were blocked by treatment with buffer (dPBS, 0.5 mM EDTA, 1% BSA) for 30 min at 22 °C. Primary antibodies (see below) were diluted in buffer and incubated with cells overnight at 4 °C, followed by two washes with FACS buffer. Secondary antibodies were diluted in buffer and incubated with cells for 1 h at 22 °C, followed by two washes with FACS buffer. Nuclei were labeled with Hoechst 33342 (Roche) diluted 1:10,000 in buffer, followed by two washes with buffer. Cells were imaged on a Leica SPE confocal microscope.

**Flow cytometry.** Cells were dissociated with Accutase, blocked with FACS buffer (dPBS, 0.5 mM EDTA, 1% BSA), and stained with fluorophore-conjugated primary antibodies (see below) according to manufacturer recommendations. Fluorescence was measured on BD FACSCanto II and BD LSRFortessa instruments, and analysis was performed with Flowing Software flow cytometry analysis suite (http://www.flowingsoftware.com).

**Antibodies.** Primary antibodies for immunoblotting: Rabbit anti-SP5 1:250 (Abcam ab209385); Mouse anti-SP1 1:500 (Santa Cruz Biotechnology sc-17824); Mouse anti-β-actin 1:1000 (Sigma A2228); Mouse anti-β-catenin 1:1000 (Sigma-Aldrich C7207). Secondary antibodies for immunoblotting: goat anti-Rabbit-IgG-HRP 1:5000 (Santa Cruz Biotechnology sc-2004); Goat anti-Mouse-IgG-HRP 1:5000 (Santa Cruz Biotechnology sc-2005). Primary antibody for IF: Rabbit anti-SP5 1:50 (Abcam ab209385), Goat anti-SOX17 1:100 (R&D Systems, AF1924). Secondary antibody for IF: Alexa Fluor 647-Goat anti-Rabbit 1:200 (Thermo Fisher Scientific A-21244); Alexa Fluor 647-Donkey anti-Mouse 1:200 (Thermo Fisher Scientific, A31571); Alexa Fluor 488-Chicken anti-Goat 1:200 (Thermo Fisher Scientific, A21467). Primary antibody for flow cytometry: APC-Mouse anti-CD184 1:200 (BD Biosciences BD555976), mouse anti-FZD7 1:100 (described in ref. [7] and available upon request from K.W.), Alexa Fluor 488 Mouse anti-Human TRA-1-81 1:100 (Biolegend, 330710); PE Mouse anti-Human SSEA4 1:100 (Biolegend, 330406). Antibodies for ChIP: Rabbit anti-SP5 polyclonal serum (custom generated by Abcam; affinity purified against a GST fusion protein with SP5 residues 1–129); Mouse anti-SP1 (Santa Cruz Biotechnology sc-17824); Rabbit anti-Trimethyl Histone H3 (Lysine 4) (EMD Millipore 04-745); Mouse anti-Dimethyl Histone H3 (Lysine 4) (EMD Millipore 05-1338); Mouse anti-Trimethyl Histone H3 (Lysine 27) (Active Motif 61017); Rabbit anti-Dimethyl Histone H3 (Lysine 27) (Cell Signaling 9728).

**Microscopy.** Phase contrast microscopy was performed using Leica, Zeiss, and EVOS (Life Technologies) inverted fluorescence microscopes. Image analysis and manipulation was performed with ImageJ and GIMP. Histological sections were imaged on a Zeiss Axioskop upright microscope. Confocal microscopy was performed using a Leica SPE microscope.

**RNA-Seq.** Total RNA was prepared from cells using the RNeasy Plus column purification kit (Qiagen) and depleted of ribosomal RNA. After ligating adaptors, fragmented RNA was converted to first strand cDNA using ArrayScript Reverse Transcriptase (Ambion), size selected (100–200 bp) by gel electrophoresis, and PCR amplified using adaptor-specific primers. Sequencing was executed on an Illumina HiSeq 2000. TopHat and Cufflinks[52, 53] were used to perform differential gene expression analysis of RNA-seq experiments. Briefly, sequencing reads were quality filtered, mapped, and aligned to the reference human genome (hg19) with TopHat and Cuffdiff was used to calculate gene expression levels as reads per thousand transcript bases per million reads mapped (RPKM). Statistically

significant changes in gene expression were obtained from RPKM values. Genes were clustered by expression pattern and principal component analysis was performed (Genesis). GO was performed (HOMER).

**ChIP-PCR, ChIP-Seq, and downstream analyses.** For SP1 and SP5 ChIP experiments, chromatin immunoprecipitations were performed according to the protocols developed by the Myers' lab (http://hudsonalpha.org/protocols; Myers Lab ChIP-Seq Protocol v041610). HESCs (H1/WA01) were treated or not treated with Wnt3a for 24 h. Formaldehyde (Sigma-Aldrich) was added to each plate to a final concentration of 1%, 10 min at room temperature. An additional plate of cells grown in parallel was used to determine cell number and an equivalent of $10^7$ cells were used per ChIP. To stop crosslinking, glycine was added to a final concentration of 0.125 M. Cross-linked cells were washed once with 1× PBS and then lysed on ice in Farnham lysis buffer (5 mM PIPES, pH 8, 85 mM KCl, 0.5% IGEPAL CA-630) (Sigma-Aldrich) freshly supplemented with and Protease Inhibitor Cocktail (Sigma-Aldrich). Cells were detached from the plate with a cell scraper, transferred to a conical tube and pelleted at 1000×g, 5 min, 4 °C. Cell pellets were resuspended in 1 mL Farnham lysis buffer supplemented with Protease Inhibitor Cocktail and pelleted at 1000×g, 5 min, 4 °C, then resuspended in 300 µL RIPA buffer (1× PBS, 1% IGEPAL CA-630, 0.5% sodium deoxycholate, 0.1 % SDS) supplemented with Protease Inhibitor Cocktail. Samples were sonicated in a Bioruptor (Diagenode), 15 min at high setting, 30 s on, 30 s off, 4 °C. Samples were centrifuged 20,000×g, 15 min, 4 °C and supernatants were flash frozen in liquid nitrogen and stored at −80 °C until immunoprecipitation. For histone modification ChIP experiments (H3K4me2, H3K4me3, H3K27me2, H3K27me3), crosslinking was performed in identical fashion but downstream processing was different: cells were lysed for 10 min on ice in lysis buffer (1% SDS, 50 mM Tris-HCl pH 8, 20 mM EDTA + Roche Complete Protease Inhibitor Cocktail). Lysates were diluted to 600 µL with TE buffer (10 mM Tris-HCl pH 8.0, 1 mM EDTA) and sonicated with a Branson Sonifier for 10 cycles (15 s on, 45 s off, 30% Power). Lysates were clarified by centrifugation (1000×g, 4 °C, 10 min) and supernatant chromatin concentrations were estimated by OD$_{260}$; 20 µL of sample was saved for input. Samples were diluted to 0.5 mg mL$^{-1}$ in dilution buffer (16.7 mM Tris-HCl pH 8, 0.01% SDS, 1.1% Triton X-100, 1.2 mM EDTA, 167 mM NaCl) and adjusted to 10% glycerol. Samples were flash frozen in liquid nitrogen and stored at −80 °C until immunoprecipitation. For each chromatin immunoprecipitation, 150 µL Dynabeads Protein G (Life Technologies, Carlsbad, CA) were resuspended by vortexing and washed three times with 1 mL PBS + bovine serum albumin (5 mg mL$^{-1}$). Beads were resuspended in a final volume of 150 µL PBS + BSA + 5 µg of antibody and incubated with rotation at 4 °C for 8 h. Beads were washed an additional three times with 1 mL PBS + BSA. About 1000 µL chromatin was adjusted with 300 µL ChIP master mix buffer (1% Triton X-100, 0.1% Deoxycholic acid, Roche Complete Protease Inhibitor Cocktail in TE buffer) and added directly to the antibody-bound beads, followed by incubation with rotation at 4 °C for at least 10 h. Supernatant was removed and beads were washed five times with 1 mL of RIPA buffer (50 mM HEPES pH 8.0, 1% IGEPAL CA-630, 0.7% Deoxycholic Acid, 500 mM LiCl, 1 mM EDTA, and Roche Complete Protease Inhibitor Cocktail) and once with 1 mL of TE buffer. About 150 µL of elution buffer (10 mM Tris-HCl pH 8, 1 mM EDTA, 1% SDS) was added directly to the beads. Beads were vortexed briefly and incubated at 65 °C for 20 min at a mixing frequency of 1300 RPM in a Thermomixer (Eppendorf). Eluted chromatin was removed from the beads and incubated at 65 °C overnight to reverse crosslinks. Chromatin was diluted with 250 µL TE buffer and treated with RNase A (0.2 mg mL$^{-1}$, 37 °C, 1 h) and Proteinase K (0.4 mg mL$^{-1}$, 55 °C, 1 h). Chromatin was purified using the Zymo ChIP Clean and Concentrator Kit (Zymo Research, Irvine, CA) and used directly for PCR or sent for library prep and sequencing. PCR primers are provided in Supplementary Table 1. Libraries from immunoprecipitated chromatin were prepared by the nucleic acid core of The Scripps Research Institute and sequenced on a HiSeq instrument (Illumina, San Diego, CA) by following the manufacturer's instruction. Reads were filtered by quality and trimmed and mapped to the genome (Bowtie 2). Random down sampling was used to normalize all tracks to have equal read number. Peaks were called (MACS[20]), BedGraph tracks were generated (BEDTools) and visualized on the Santa Cruz Genome Browser. Motif, coverage, annotation, and GO analyses were performed (HOMER). Venn diagrams were generated (BioVenn).

**Data availability.** The RNA-seq and ChIP-seq data discussed in this publication have been deposited in NCBI's Gene Expression Omnibus and are accessible through GEO Series accession number GSE103175. All data pertinent to this study are available in Supplementary Information. Additional information on these data is available from the corresponding authors.

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

## Acknowledgements

We are grateful to Dr. Bing Ren for advice and Ah Young Lee for technical support of ChIP-Seq experiments, Dr. Rick Myers and Kim Newberry for advice on the SP1 ChIP-Seq experiment, Dr. Chad Cowan (Harvard Stem Cell Institute, Cambridge, MA) for kindly providing Cas9 and guide RNA plasmids, and to Fumitaka Osakada for pCMMP_IRES-GFP plasmid. We are grateful to Steve Head, Lana Schaffer, and John Shimashita of the Next Generation Sequencing Core (The Scripps Research Institute, La Jolla, CA) for performing sequencing reactions, Cleber Trujillo, and Nicholas Liang for assistance with teratoma assays, and to Jesus Olvera and Cody Fine of the UCSD Human Embryonic Stem Cell Core Facility for technical assistance of flow cytometry experiments. I.J.H. was supported in part by the UCSD Interdisciplinary Stem Cell

Training Program (CIRM TG2-01154), and A.P. was supported by California State University-San Marcos CIRM Bridges to the Stem Cell Research Training Grant (CIRM TB1-01186). This work was supported by grants to K.W. from the California Institute for Regenerative Medicine (CIRM, RB1-01406), the National Institute of Health (NIH, 1R01GM110304-01) and by the UCSD Stem Cell Program, and was made possible in part by the CIRM Major Facilities grant (FA1-00607) to the Sanford Consortium for Regenerative Medicine. The content of this manuscript is solely the responsibility of the authors and does not necessarily represent the official views of the NIH or CIRM.

## Author contributions

Conceptualization: K.W. and T.G.; Methodology: I.J.H., K.W., and T.G.; Investigation: I.J.H., T.B., O.G., B.L., A.P., C.J., C.R., J.R. and D.B.; Writing original draft: I.J.H. and K.W.; Writing, review, and editing, I.J.H., D.B., T.G. and K.W. Funding and acquisition: I.J.H., T.G. and K.W.; Resources: T.G. and K.W.; Supervision: T.G. and K.W.

## Additional information

**Competing interests:** The authors declare no competing financial interests.

