## [Peer Review File · Nature Communications]

Reviewers' comments:

Reviewer #1 (Remarks to the Author):

The manuscript "The WNT target SP5 silences expression of WNT promoted transcription programs" by Huggin et al conducted RNA sequencing assays after treatment of hESCs with Wnt3a, and found that SP5 was most significantly induced. Then, the function of SP5 was studied through both loss and gain function strategies and authors proposed that SP5 restricts endoderm formation. Subsequently, authors carried out SP5 ChIP-seq assay along with that of SP1 in cells treated with Wnt3a. Combined with gene expression data, authors concluded that SP5 functions to downregulate expression of genes that are activated by WNT signaling. Finally, authors proposed that SP5 is a Wnt target gene in cells with developmental potential. Overall, the manuscript is clearly written. However, there is a lack of sufficient experimental data to fully support their conclusions. The study does not present enough novelty and impact to further our understanding how SP5 contributes to the function of Wnt signaling.

Major concerns:

1. The novelty of finding is not strong. It has been already reported that SP5 is a downstream target of Wnt signaling, acting as either a transcriptional repressor or activator. Moreover, the relationship between SP1 and SP5 downstream of Wnt signaling described here is also known. In particular, a recently published study has reported SP5 DNA binding profiles in mouse embryos and differentiating ESCs.
2. The evidence that SP5 restricts endoderm formation is not solid. Data presented in figure 2f, 2g and Supplemental Fig 2i are not convincing to make conclusion that SP5 acts to regulate the endoderm formation. It seems that there is a problem in analyzing data of Fig 2g and Supplemental Fig 2i.
3. Results in Fig 5 are most descriptive, lacking molecular insights and functional consequences.

Reviewer #2 (Remarks to the Author):

Signaling pathways generally integrate multiple pathway components to provide negative feedback mechanisms at various steps of pathway activation. Mechanisms underlying downregulation/termination of transcriptional programs activated by the signaling pathway are, however, less well understood. Huggins et al. use human pluripotent stem cells to study transcriptional response following activation of the Wnt pathway. The authors specifically focus on the role of transcriptional repressor SP5, a well-known target gene of canonical Wnt signaling, as a key molecular player to downregulate genes initially activated by canonical Wnt signaling. In the very-well designed and executed study the authors use a combination of transcriptomic and epigenetic profiling in wild type and SP5-deficient cells to identify targets of SP5 upon Wnt pathway activation. In order to test the hypothesis of Fujimura et al. that SP5 inhibits SP1 target genes authors performed ChIP-seq for SP5 and SP1 in parallel. Surprisingly, a rather restricted set of SP5 binding sites (662) was identified in the entire genome under conditions used – a significant number of those SP5 binding sites being in genes ectopically activated by Wnt signaling in SP5-deficient hPSC. This remarkable result suggest that SP5 specifically contributes to downregulation of transcriptional targets of Wnt signaling. Given the generality of the mechanism proposed here it is hard to explain the difference between function of Sp5 in mouse (Kennedy et al. 2016) and human system as well as the small overlap in the SP5 genomic targets.

The study is straightforward and technically sound. It provides a large amount of data supporting the role of SP5 in downregulation of Wnt transcriptional program. Some questions, however, remain unanswered.

First of all, the mechanism for restricted targeting of SP5 (662 SP5 binding sites compared to 24,747 SP1 sites) was not investigated. How Sp5 is recruited to only a limited set of genomic targets significant number of which are Wnt/b-catenin signaling target genes? Given the consensus/logo being shared by SP5 and SP1 the targeting process has to be highly regulated. Kennedy et al. show that Sp5 interacts with Tcf/Lef. Does Sp5 bind to Tcf/Lef in Wnt3a-stimulated hPSC to be recruited near Wnt-target genes? Authors have anti-SP5 antibody, so they might be able to perform time-course co-immunoprecipitations in Wnt3a-stimulated hPSC.

SP5 promoter is a target of SP5 protein upon Wnt3A pathway stimulation at 24h. Given the mechanism proposed here as a main outcome of the paper it follows that SP5 repressor should start to diminish SP5 gene expression. Yet, at 48 hours transcriptomics data show an increase of SP5 transcription in WT cells (Fig1f). Authors should extend the time window of transcriptional profiling in order to see downregulation of SP5 gene later on during continuous (or pulse) Wnt3A stimulation. Isn't this the way to show the termination phase for the model shown in Fig.6 by experimental means?

Minor comments.

The core of the paper is the ChIP-seq using a novel proprietary SP5 antibody generated against N-terminus of SP5. This region contains a short stretch of homology (aa31-about aa50) to other zinc finger proteins (peptide SPLA...PG). It makes sense to provide a small characterization panel (western, IP) for this essential tool as a supplementary information. Please state consistently if aa1-129 (Methods) or aa1-133 (Supplementary Fig.1f) is correct. Also, instead of stating in Methods as a source of antibody (line 490) ...provided by Abcam... it is more appropriate to saycustom generated by Abcam.

The authors made a considerable effort on investigating the mechanism for an interesting observation that SP5 is significantly more Wnt-inducible in stem-like cells and during differentiation the SP5 ability to be induced by Wnt signalling diminishes. Authors conclude that changes in poised chromatin likely accounts for the loss of SP5 induction in response to Wnt3A. Why were H3K27me2 and H3K4me2 marks used for ChIP-PCR in Sp5 promoter region(Fig 5)? H3K4me1 and H3K4me2 are enriched at enhancers and H3K4me3 is enriched at promoters (reviewed in Chen and Dent), so maybe H3K4me3 should be checked. Since H3K27me3 surprisingly showed non-significant change during differentiation H3K9me3 that also marks facultative heterochromatin should be examined.

Why is the TSS+500 window not the same scale for all genes in Supp.Fig1b?

Perhaps it makes more sense to have both loss- (shSP5-GFP) and gain-of function (SP5-GFP) experiment shown together in the supplement or main Figure. In left panel of Fig2g the green and grey bar appear to be swapped based on numbers in the left panel.

Indicate what red dots (more stem-like) and grey dots stand for in Fig5a.

Heat map shows that other Klf/SP genes (Klf7, Klf8 and even SP8) are slightly induced during Wnt3A stimulation albeit their absolute levels (RPKM) are much lower than that of SP5. What is the induction of the respective genes between 0h and 48 hours?

Line 118.....SP5 mutant cells expressed significantly higher levels of the DE markers SOX17 and CXCR4 compared to WT cells (Fig. 2f)..... well yes, but not at all stages, please re-phrase

Reviewer #3 (Remarks to the Author):

Huggins and colleagues describe a novel negative feedback loop mechanism terminating Wnt/ β -catenin-driven transcription in pluripotent cells, in particular in hPSCs. In contrast to known negative feedback systems attenuating Wnt outputs at the level of ligand, receptor complexes or the intracellular Wnt-cascade, the mechanism determined by Huggins and colleagues is driven by the Wnt-target gene SP5. In hPSCs, SP5 accumulates during continuous Wnt stimulation and competes with ubiquitous SP1 transcription factor for binding sites in the vicinity of Wnt-activated genes. Thereby, SP5 binding acts to diminish and terminate the transcriptional program previously activated by Wnt signaling. Such competition between SP5 and SP1 was suggested previously (Fujimura et al., 2007), but the authors report new and completely unexpected facets, thereby completing the complex mosaic of Wnt-termination mechanisms. Mainly, the mode of SP5 action characterized by the authors points towards a new possible regulatory node or switch controlling the differentiation process(es) of pluripotent cells. In general, the manuscript is of high impact with solid data of significant importance. I recommend the manuscript for publication in Nature Communications. However, there are some critical concerns that need to be addressed by authors prior to publication.

Criticism and question (not sorted according to their importance):

Fig.1a (and others): The authors used 1nM Wnt3a. How was this concentration calculated, i.e. the molecular weight determined of Wnt3a? Functional Wnt protein is heavily post-translationally modified (with possibly complex glycosylation pattern).

Fig.1a (and other Figs.) The changes in morphology may not necessarily reflect the pluripotent/differentiated status of stem cells (reported for example in β -catenin-null mESCs that look as differentiated, but still express key stem cell markers). Although hPSCs are of different origin (and stage) it would be worthwhile to show immunofluorescent staining (stem cells vs. differentiation markers and/or some targets/GOIs). Immunostaining would also increase the impact of other figures (Fig. 2d,e,f and Fig.4 a,b), since changes in mRNA level may not fully reflect protein levels.

Fig.1b and Fig.4a would benefit by indicating directly within the figure(s) or at least in the figure legend what particular classes/groups of genes (i - iv) are represented; this would facilitate the reading.

Fig1e Can the authors show a higher magnification (i.e. to show the detail/inset in higher magnification)? In some cells it is hard to see if the SP5 is really nuclear.

Fig2b Could be moved (especially the lower panel) to a supplementary figure.

Fig.2e,f If mutating SP5 pushes hPSCs cells into endodermal (DE) differentiation, are the other branches of differentiation affected in dZF mutants (e.g. differentiation into neuroectoderm etc.)? Did author check this?

Fig.2f (and main text lines 118 - 121). It is mention within the text:..." SP5 mutant cells expressed significantly higher levels of the DE markers SOX17 and CXCR4 compared to WT cells (Fig. 2f). In contrast, expression of the mesendodermal marker T was similar in WT and dZF cells...". Although the differences in expression the data seems to be significant, due to very consistent levels as apparent based on the low error bars showing SEM or SD (please indicate), the significance reduces over time and the variations are, in general, not so dramatic. For instance, CXCR4 in dZF1 cells is expressed at the same level as in wt cells after 3 days (end of differentiation protocol). On the other hand, T is expressed significantly lower at 2 days. This is the time point, when also the differences in expression of other targets are most prominent. Can the authors comment on this? Or change the text to be more precise?

Fig.2g The more than 10% difference in GFP(-) cells between shControl and shSP5 is close to that one observed in the middle panel (14,9%). Should the percentage of GFP(-) cells expressing gated CXCR4 level not be more comparable between experiments? If it is so variable can the authors perform the following experiment: to use different labeling for shControl and shSP5 (e.g. red and green) mix the cells together and compare really shControl and shSP5 within one experiment?

Fig.4a The class/group (iv) of the Wnt-target genes is most strongly upregulated after 48 hours in both wt and dZF cells (however in dZF1 it is higher). If SP5 accumulates over time and starts to repress the transcription of target genes should the level of transcripts not be lower in wt (and not highest) at the last time point (differentiated cells)? Or it is just a matter of duration and later it will decrease in wt cells (and not in mutants)? Can authors comment on this?

Fig. 4b, e If dZF mutants express SP5 that cannot bind genomic DNA (did the authors test this?) what happens at the regions of SP5-dependent genes, at loci where SP5 peaks are present in wt cells? If SP5 can not bind there (and thus outcompete SP1) the regulatory site may remain free for SP1. Is SP1 bound there (i.e. as in wt unstimulated cells). It would be worthwhile to show SP1/SP5 ChIP for some targets (e.g. T, SP5, TBX3...). At least the authors should comment on this. (TCF or b-catenin ChIP might be also very interesting to show).

Fig.5 c The authors claim that the ratio of AXIN2 in Wnt3a-treated vs. unstimulated cells changed only slightly (main text line 227-228). In fact, 28 days after treatment the expression of AXIN2 is insensitive to Wnt3a stimulation comparable to SP5. What is different is the dynamics of the process ("sensitivity" of SP5 decreases during time, AXIN2 goes up and decreases afterwards). Can the authors comment on this (or change the text accordingly)?

Can the authors speculate more (or, better, can they check) about the nature of this Wnt-desensitisation? For example: are some secreted inhibitors expressed (sFRPs, Dkk), what is the nature of TCF/LEFs (in mESCs a TCF/LEF switch was described), is Tankyrase more active.....

Suppl.Fig.1b It would be worthwhile showing a better quality image to recognize peaks more clearly.

Suppl.Fig. 6a It would be very helpful to indicate the origin / type of cell line that was tested (as an additional column within the table).

Text line 315 hiPSCs are mentioned to have been tested. However, they are not indicated in any figure. If the authors have data based on hiPSCs it would be worthwhile to show them. The data would increase the impact of the manuscript beyond hPSCs.

Other comments:

The title may not really reflect what the paper shows; would it not be more accurate to use a variation of the running title?

The abbreviation for the hPSCs should appear in text line 4 for the first time and not in line 38.

Text line 51: The duration and method (recombinant protein, plasmid?) of the treatment should be mentioned.

REBUTTAL TO REVIEWER COMMENTS

Reviewer comments are italicized, responses are not.

Reviewer #1 (Remarks to the Author):

1. The novelty of finding is not strong. It has been already reported that SP5 is a downstream target of Wnt signaling, acting as either a transcriptional repressor or activator. Moreover, the relationship between SP1 and SP5 downstream of Wnt signaling described here is also known. In particular, a recently published study has reported SP5 DNA binding profiles in mouse embryos and differentiating ESCs.

RESPONSE

Our study focuses on a system in which SP5 is found to be exclusively a repressor. Further, it is found here for the first time to be highly selective in where it binds DNA to impose that repression. The prior work noted by this reviewer laid the groundwork for understanding SP5's importance but left open major ambiguities, including whether it is a repressor or activator or both, under what conditions, and of what genes, and what is the critical outcome of the repression. The results reported here eliminate the prior ambiguities and underscore the critical importance of SP5 as a negative regulator of the Wnt signaling pathway at the level of its targets, rather than at the level of pathway components. Thus, although the reviewer raises several important and valid points of criticism, we respectfully disagree with the assertion that the novelty of these findings is not strong.

To our knowledge, our comprehensive study is the first of its kind to perform and analyze a time series of signaling in human pluripotent stem cells. We integrate genome-wide transcriptional and chromatin data to identify a previously unknown mechanism by which Wnt, a key developmental growth factor, regulates developmental programs. The model that emerges from our studies, namely that SP5 acts to negatively regulate expression of the entire Wnt transcriptional program, is a conceptual advance that is of significant interest and importance to the community of developmental and stem cell biologists. The concept of developmental competence has been the subject of many years of intense research in developmental biology. Our study contributes to this field by demonstrating that mechanisms exist that dampen the transcriptional response to developmental regulators, in this case Wnt. Such dampening of transcriptional programs is arguably equally as important in differentiation as the initial establishment of developmental competence. The reviewer is correct that other studies have examined the role of SP5 in Wnt signaling. Our study builds on these prior studies and puts forth a model that has not been proposed previously. We further contend that our model will have implications beyond Wnt signaling, as other developmental signaling pathways may be regulated by a similar mechanism.

2. The evidence that SP5 restricts endoderm formation is not solid. Data presented in figure 2f, 2g and Supplemental Fig 2i are not convincing to make conclusion that SP5 acts to regulate the endoderm formation. It seems that there is a problem in analyzing data of Fig 2g and Supplemental Fig 2i.

RESPONSE

What is indisputable is that SP5 regulates differentiation, and the findings here report this for the first time. This comment from the reviewer helped us realize that Fig. 2g and Supplementary Fig. 2i (Supplementary Fig. 2i and j in the revised version) were laid out in a complicated manner that made the data difficult to follow. The effects on

endodermal differentiation are quite subtle, and that contributed to the difficulty in reading/following the figure. To clarify the paper and highlight the most important point, that SP5 indeed regulates differentiation, we have focused our conclusions and interpretations of the Figure 2 experiments on this point.

Our data demonstrates solidly that SP5 regulates differentiation. To draw this conclusion, we now present several independent pieces of data: (1) the transcriptional response of SP5 mutant cells is distinct from that of WT cells; (2) SP5 mutant cells exhibit a differentiation defect compared to WT cells; (3) SP5 mutant cells show defects even during directed differentiation; (4) shRNA-mediated knockdown of SP5 also shows differentiation defect; and (5) SP5 overexpression diminishes differentiation, shown here for the case of endoderm. Together, we respectfully submit that our central conclusion that SP5 regulates hPSC differentiation, is warranted and well-supported.

With respect to analysis of data in Fig. 2g and Supplementary Fig. 2i (now consolidated into Supplementary Fig. 2i and j), we direct this reviewer to our response to comment 8 by reviewer 3.

3. Results in Fig 5 are most descriptive, lacking molecular insights and functional consequences.

RESPONSE

The purpose of Figure 5 is to delineate the stem cell context of SP5's role as a post-hoc negative regulator of Wnt signaling – post-hoc in the sense that it dampens the up-regulated targets of Wnt signaling in a time-sensitive manner. The reviewer is correct in noting the results in Figure 5 are descriptive, but that is the goal: to characterize the stem cell context in which SP5 has an essential role. Although we do not prove that SP5 is a stem cell-specific Wnt target gene—and we make no such bold claim—we provide substantial evidence that induction of SP5 in response to Wnt signaling correlates with stemness properties. This observation will be of interest to the scientific community and help guide others as they study SP5 in their own systems. The publication of the correlation between SP5 induction and stemness as part of this study is of value. Several other studies and reviews have suggested that SP5, like AXIN2, is a universal Wnt target gene. Our studies challenge this claim and indicate that SP5 expression in response to Wnt signaling is more selective than that of AXIN2. These findings, despite their descriptive nature, will be of significant interest to researchers studying developmental processes that are affected by Wnt signaling. For example, while Axin2 is commonly used as an in vivo reporter for Wnt signaling and for lineage tracing experiments in mice, similar experiments with Sp5 will likely yield significant differences. Therefore, our observation that SP5 expression in response to Wnt signaling is highly cell context dependent represents an important and valuable contribution.

Reviewer #2 (Remarks to the Author):

QUESTION/COMMENT

1. First of all, the mechanism for restricted targeting of SP5 (662 SP5 binding sites compared to 24,747 SP1 sites) was not investigated. How Sp5 is recruited to only a limited set of genomic targets significant number of which are Wnt/b-catenin signaling target genes? Given the consensus/logo being shared by SP5 and SP1 the targeting process has to be highly regulated. Kennedy et al. show that Sp5 interacts with Tcf/Lef. Does Sp5 bind to Tcf/Lef in Wnt3a-stimulated hPSC to be recruited near Wnt-target

genes? Authors have anti-SP5 antibody, so they might be able to perform time-course co-immunoprecipitations in Wnt3a-stimulated hPSC.

RESPONSE

This is indeed a critical point that we had not addressed in the original version of the paper. Here we provide additional data that sheds light on the likely mechanism by which SP5 binding is highly selective compared to SP1, a ubiquitously acting transcription factor. The observation that SP5 targets WNT target genes and subsequently represses their expression suggested that SP5 interacts with core components that mediate the transcriptional response to a WNT signal, specifically beta-catenin and/or LEF/TCF. To test this possibility, we used a proximity ligation method called APEX2 (Lam et al., Nat Methods 12: 51-4, 2015, PMID: 25419960), which uses reactive biotin to label proteins within a 10nm radius of the APEX2-tagged protein. Using this method we find that SP5-APEX2 leads to labeling of beta-catenin, providing strong evidence that these two proteins are in close proximity. As a control for specificity of this highly specific labeling method, we show that the abundant protein beta-actin is not labeled. We have incorporated this data in the revised manuscript in Supplementary Figure 4c.

Of note, we attempted to show this interaction by co-immunoprecipitation of endogenous SP5 and beta-catenin, however, we were unable to do so. We attribute this to the fact that SP5 is a nuclear protein and conditions required to detect endogenous SP5 require harsh lysis conditions that disrupt most protein-protein interactions. These harsh lysis conditions are not problematic for the ChIP-seq experiments described in our paper since cells are cross-linked with paraformaldehyde, which stabilizes the SP5-DNA complexes.

In addition, it should also be noted that a previous publication provided evidence for Sp5 and beta-catenin binding (Kennedy et al, PNAS, PMID: 26969725), further substantiating our claim for this interaction.

2. SP5 promoter is a target of SP5 protein upon Wnt3A pathway stimulation at 24h. Given the mechanism proposed here as a main outcome of the paper it follows that SP5 repressor should start to diminish SP5 gene expression. Yet, at 48 hours transcriptomics data show an increase of SP5 transcription in WT cells (Fig1f). Authors should extend the time window of transcriptional profiling in order to see downregulation of SP5 gene later on during continuous (or pulse) Wnt3A stimulation. Isn't this the way to show the termination phase for the model shown in Fig.6 by experimental means?

RESPONSE

This is an excellent point. We have performed a longer time-course of Wnt3a treatment and observe a robust down-regulation after 4 days of treatment. Furthermore, and very importantly, this down-regulation of SP5 is not observed in SP5 mutant cells. We have included this data in Figure 4f with the associated text in the manuscript. Together these data provide strong evidence that SP5 feeds back to repress expression of Wnt target genes, including itself.

Minor comments.

3. The core of the paper is the ChIP-seq using a novel proprietary SP5 antibody generated against N-terminus of SP5. This region contains a short stretch of homology (aa31-about aa50) to other zinc finger proteins (peptide SPLA...PG). It makes sense to provide a small characterization panel (western, IP) for this essential tool as a supplementary information. Please state consistently if aa1-129 (Methods) or aa1-133

(Supplementary Fig. 1f) is correct. Also, instead of stating in Methods as a source of antibody (line 490) ...provided by Abcam... it is more appropriate to saycustom generated by Abcam.

RESPONSE

The SP5 antibody used in these studies has recently become available through Abcam (Anti-Sp5 antibody, ab209385). The exact identity of the immunogen used to generate this antibody was never revealed to us and the sequence remains proprietary, as indicated in the data sheet. The now-commercially-available monoclonal antibody was used for Western blotting and immuno-fluorescence (Figures 1d, 1e, 2b).

Prior to the isolation of this monoclonal antibody, we obtained small samples of crude serum from rabbits that had been immunized with the proprietary N-terminal synthetic peptide. Knowing that the antibody was raised to some portion of the N-terminus of SP5, we generated a GST fusion protein with amino acids 1-129 of SP5, and used this fusion protein to affinity purify the crude antiserum. The affinity-purified antiserum was then used for the ChIP-seq experiments. Given the extremely time consuming and costly process for ChIP-seq experiments, we respectfully submit that repeating these experiments with the new SP5 monoclonal antibody would be excessive. Most importantly, such repeat experiments will not alter the conclusions of our manuscript. We have corrected Supplementary Fig. 1f to indicate aa1-129.

4. The authors made a considerable effort on investigating the mechanism for an interesting observation that SP5 is significantly more Wnt-inducible in stem-like cells and during differentiation the SP5 ability to be induced by Wnt signalling diminishes. Authors conclude that changes in poised chromatin likely accounts for the loss of SP5 induction in response to Wnt3A. Why were H3K27me2 and H3K4me2 marks used for ChIP-PCR in Sp5 promoter region(Fig 5)? H3K4me1 and H3K4me2 are enriched at enhancers and H3K4me3 is enriched at promoters (reviewed in Chen and Dent), so maybe H3K4me3 should be checked. Since H3K27me3 surprisingly showed non-significant change during differentiation H3K9me3 that also marks facultative heterochromatin should be examined.

RESPONSE

The mechanism by which the SP5 promoter is silenced in differentiated cells relative to undifferentiated cells is certainly of interest, and we provide additional data as requested by this reviewer. Specifically, we show that the promoter mark H3K4me3 is highly enriched in the SP5 promoter in undifferentiated hES cells compared to Day 27 differentiated cells this data in Figure 5d. The data for the chromatin mark H3K27me2, previously in Figure 5d, has been moved to Supplementary Figure 6e.

5. Why is the TSS+500 window not the same scale for all genes in Supp.Fig1b?

RESPONSE

The reason for this was that this histone mark as visualized using these wiggle plots varied substantially from gene to gene. However, we agree with the reviewer that this makes it difficult to look at and we have adjusted the figure such that the scale is identical for all genes shown. The important point that this histone mark is enriched upon Wnt3a treatment at these loci is clearly visible.

6. Perhaps it makes more sense to have both loss- (shSP5-GFP) and gain-of function (SP5-GFP) experiment shown together in the supplement or main Figure. In left panel of

Fig2g the green and grey bar appear to be swapped based on numbers in the left panel.

RESPONSE

We agree that showing this data together will make it clearer. We therefore consolidated the loss- and gain-of-function experiments in Supplementary Figure 2i and j.

7. Indicate what red dots (more stem-like) and grey dots stand for in Fig5a.

RESPONSE

We have modified this figure to make the point more clearly, and we have added additional text to the corresponding figure legend.

8. Heat map shows that other Klf/SP genes (Klf7, Klf8 and even SP8) are slightly induced during Wnt3A stimulation albeit their absolute levels (RPKM) are much lower than that of SP5. What is the induction of the respective genes between 0h and 48 hours?

RESPONSE

The reviewer is correct that these members of the KLF/SP family are upregulated. However, the fold induction is significantly lower than that of SP5. The RPKMs from our RNA-seq experiment, which is provided in Supplementary Table 1, details levels of induction.

Gene ID	Hours Wnt3a				Fold induction (min vs max)
	0	12	24	48	
SP8	3.2	5.1	5.2	9.9	3.1
KLF7	9.5	11.5	16.3	17.3	1.8
KLF8	5.0	4.0	4.7	13.3	2.7

9. Line 118.....SP5 mutant cells expressed significantly higher levels of the DE markers SOX17 and CXCR4 compared to WT cells (Fig. 2f)..... well yes, but not at all stages, please re-phrase

RESPONSE

The text has been modified to indicate that expression of DE markers in SP5 mutant cells is elevated relative to WT cells.

Reviewer #3 (Remarks to the Author):

Criticism and question (not sorted according to their importance):

1. Fig.1a (and others): The authors used 1nM Wnt3a. How was this concentration calculated, i.e. the molecular weight determined of Wnt3a? Functional Wnt protein is heavily post-translationally modified (with possibly complex glycosylation pattern).

RESPONSE

We generate our recombinant Wnt proteins in house using methods that we have

developed and optimized (e.g. see PMID: 19099243), and determine the concentration of each batch. To get an approximate value for the concentration of a recombinant Wnt3a protein batch, we visualize the purified protein on a Coomassie-stained gel. Using a dilution series of a protein standard (e.g. BSA from 10 ng to 1ug), we are able to estimate the concentration of Wnt3a in ug/ml. Wnt3a migrates as single band—albeit somewhat fuzzy band which is likely due to variable amounts of glycosylation—at ~40kDa. Using these values for MW and concentration, we convert ug/ml to molarity. In general, our Wnt3a stock solution is at a concentration of 1-5 uM (40ug/ml – 200ug/ml).

2. Fig. 1a (and other Figs.) The changes in morphology may not necessarily reflect the pluripotent/differentiated status of stem cells (reported for example in b-catenin-null mESCs that look as differentiated, but still express key stem cell markers). Although hPSCs are of different origin (and stage) it would be worthwhile to show immunofluorescent staining (stem cells vs. differentiation markers and/or some targets/GOIs). Immunostaining would also increase the impact of other figures (Fig. 2d,e,f and Fig. 4 a,b), since changes in mRNA level may not fully reflect protein levels.

RESPONSE

We agree with the reviewer that the morphological changes in response to Wnt3a treatment are subtle. We therefore included additional data to show by immunofluorescence and flow cytometry both the acquisition of differentiation markers (SOX17 in Figure 1b) and the loss of pluripotency markers (FZD7, SSEA4 and TRA1-81 in Supplementary Figure 1a and b).

3. Fig. 1b and Fig. 4a would benefit by indicating directly within the figure(s) or at least in the figure legend what particular classes/groups of genes (i – iv) are represented; this would facilitate the reading.

RESPONSE

We have added these descriptions to the figures (Figure 1c and 4a) to clarify the individual gene sets.

4. Fig1e Can the authors show a higher magnification (i.e. to show the detail/inset in higher magnification)? In some cells it is hard to see if the SP5 is really nuclear.

RESPONSE

We have included higher resolution images and a higher magnification of the inset to clearly show that SP5 is primarily localized to the nucleus.

5. Fig2b Could be moved (especially the lower panel) to a supplementary figure.

RESPONSE

Thank you for this suggestion, however, we feel that showing protein levels and the longer exposure of the same immunoblot is important, as it demonstrates that very little truncated SP5 protein is present in the mutant cells. Such truncated proteins, if abundant, could exert dominant effects. Given the low amounts of these truncated proteins, which is clearly documented in this figure, we contend that the SP5 mutation is a complete loss of function mutation.

6. Fig. 2e,f If mutating SP5 pushes hPSCs cells into endodermal (DE) differentiation, are the other branches of differentiation affected in dZF mutants (e.g. differentiation into

neuroectoderm etc.)? Did author check this?

RESPONSE

The main point of the data in this figure is to show that differentiation of SP5 mutant hESCs is impaired compared to wildtype hESCs. Consistent with our model we observe a significant shift towards mesendodermal differentiation. In this revised version we provide additional data (Figure 2d) that this mesendodermal skewing is accompanied with a concomitant loss of ectodermal differentiation, as highlighted by a significant elevation of PAX6 expression in WT cells compared to SP5-mutant cells.

7. Fig.2f (and main text lines 118 – 121). It is mention within the text:...” SP5 mutant cells expressed significantly higher levels of the DE markers SOX17 and CXCR4 compared to WT cells (Fig. 2f). In contrast, expression of the mesendodermal marker T was similar in WT and dZF cells...”.

Although the differences in expression the data seems to be significant, due to very consistent levels as apparent based on the low error bars showing SEM or SD (please indicate), the significance reduces over time and the variations are, in general, not so dramatic. For instance, CXCR4 in dZF1 cells is expressed at the same level as in wt cells after 3 days (end of differentiation protocol). On the other hand, T is expressed significantly lower at 2 days. This is the time point, when also the differences in expression of other targets are most prominent. Can the authors comment on this? Or change the text to be more precise?

RESPONSE

We agree with the reviewer that the effects of SP5 knockout (KO) on endodermal differentiation are subtle, especially at later time points. In the revised manuscript, we address this point, emphasizing that SP5 KO significantly alters the differentiation potential of hPSCs. While there is a clear skewing towards mesendodermal lineages during non-directed differentiation and towards DE during directed differentiation, the primary effect we wish to highlight is the difference in differentiation potential between WT and SP5 mutant cells.

Error bars are SEM for 4 technical replicates.

8. Fig.2g The more than 10% difference in GFP(-) cells between shControl and shSP5 is close to that one observed in the middle panel (14,9%). Should the percentage of GFP(-) cells expressing gated CXCR4 level not be more comparable between experiments? If it is so variable can the authors perform the following experiment: to use different labeling for shControl and shSP5 (e.g. red and green) mix the cells together and compare really shControl and shSP5 within one experiment?

RESPONSE

The efficiency of hPSC differentiation into DE is highly variable and dependent on various factors, such as seeding density and cell line. The slight differences in DE formation between the various differentiations, ranging from 40.1 to 64.1% (Supplementary Figure 2i and j), do not influence the interpretation of the data. For each experiment—shRNA-mediated knockdown (KD) and overexpression (OE)—the important difference is between the GFP-positive population (= transduced cells) relative to the GFP-negative population (= non-transduced cells). In the case of KD (Supplementary Figure 2i), CXCR4 expression is elevated in the GFP-positive population of SP5-shRNA transduced cells, an effect not observed for the control shRNA transduced cells. Conversely, OE of SP5 produces the opposite effect, i.e. a decline in

CXCR4 expressing cells in SP5 overexpressing population (Supplementary Figure 2j). Given the reproducibility of this experiment (we have performed this experiment 3 times with the same result), the complementary results for KD and OE, and the similar effects observed for dZF cells, we feel that these experiments convincingly demonstrate that SP5 influences the differentiation potential of hPSCs.

9. Fig. 4a The class/group (iv) of the Wnt-target genes is most strongly upregulated after 48 hours in both wt and dZF cells (however in dZF1 it is higher). If SP5 accumulates over time and starts to repress the transcription of target genes should the level of transcripts not be lower in wt (and not highest) at the last time point (differentiated cells)? Or it is just a matter of duration and later it will decrease in wt cells (and not in mutants)? Can authors comment on this?

RESPONSE

We appreciate this astute observation. Gene expression in response to Wnt3a is highly dynamic, in part because of the negative feedback exerted by SP5, which we describe in this manuscript. In the SP5 mutant cells, the normal brakes that restrain expression of Wnt target genes are released and expression of these genes increases compared to WT cells. Furthermore, it follows, as pointed out by the reviewer, that in WT cells expression of these Wnt targets should decline as cells proceed through differentiation. To address this point, we performed a longer time course of Wnt3a treatment and found that expression of SP5 declines after 3 days, consistent with negative feedback regulation. Furthermore, this decline is not observed in SP5 mutant cells, further supporting the idea that the normal brakes are lacking. We have included this new data in Figure 4f.

10a. Fig. 4b, e If dZF mutants express SP5 that cannot bind genomic DNA (did the authors test this?)

RESPONSE

We have not tested whether the mutant SP5 protein binds genomic DNA, however, given that the entire zinc finger domain is deleted and very little mutant protein is produced, we find it highly unlikely that this truncated protein has any DNA binding activity.

10b. Fig. 4b, e If dZF mutants express SP5 that cannot bind genomic DNA (did the authors test this?) what happens at the regions of SP5-dependent genes, at loci where SP5 peaks are present in wt cells? If SP5 can not bind there (and thus outcompete SP1) the regulatory site may remain free for SP1. Is SP1 bound there (i.e. as in wt unstimulated cells). It would be worthwhile to show SP1/SP5 ChIP for some targets (e.g. T, SP5, TBX3...). At least the authors should comment on this. (TCF or b-catenin ChIP might be also very interesting to show).

RESPONSE

This is in fact an interesting experiment, one that we have attempted. We performed SP1 ChIP from WT and dZF cells followed by qPCR for the promoter regions of *SP5* and *T*. Results are unfortunately inconsistent and difficult to explain. A representative experiment is shown below. The increased SP1 occupancy at the *SP5* promoter in dZF cells is potentially interesting, however, we do not observe this effect for the *T* promoter. One potential explanation for the effect at the *SP5* promoter may be that CRISPR/Cas9-mediated deletion of a large portion of the *SP5* gene may have altered the local

chromatin structure at this locus. Overall, these experiments are inconclusive and fail to provide additional insights into the competition between SP1 and SP5. In light of our many other results (including Figure 3f and Supplementary Figure 3a), as well as those of others (e.g. Fujimura et al. J Biol Chem 282: 1225-37, 2007), we respectfully submit that the conclusion of SP1-SP5 competition at select sites has strong support.

11. *Fig.5 c The authors claim that the ratio of AXIN2 in Wnt3a-treated vs. unstimulated cells changed only slightly (main text line 227-228). In fact, 28 days after treatment the expression of AXIN2 is insensitive to Wnt3a stimulation comparable to SP5. What is different is the dynamics of the process (“sensitivity” of SP5 decreases during time, AXIN2 goes up and decreases afterwards). Can the authors comment on this (or change the text accordingly)?*

RESPONSE

The reviewer is correct in stating that expression of AXIN2 in response to Wnt3a is comparable to that observed for SP5 in differentiated cells. The main point of this experiment is to show the desensitization in SP5 expression in response to Wnt3a; for these experiments, AXIN2 served as a “control” since it is considered to be a universal Wnt target gene. In contrast to SP5, AXIN2’s inducibility is only marginally affected. We have rephrased the text such that it more accurately describes the AXIN2 data.

12. *Can the authors speculate more (or, better, can they check) about the nature of this Wnt-desensitisation? For example: are some secreted inhibitors expressed (sFRPs, Dkk), what is the nature of TCF/LEFs (in mESCs a TCF/LEF switch was described), is Tankyrase more active.....*

RESPONSE

Our data indicate that SP5 induction in response to Wnt is silenced through loss of epigenetic marks that are associated with active promoters, specifically H3K4me2 and H3K4me3, as shown in Figure 5e. However, other mechanisms, such as those suggested by the reviewer may be at play here as well. We performed the following experiment to further address this possibility: rather than stimulate the pathway with Wnt3a, we used a GSK3 inhibitor, CHIR98014, which acts downstream of Wnts and their receptors. Like Wnt3a, CHIR failed to potently activate SP5 transcription in 28-day differentiated cells, indicating that this de-sensitization is not due to reduced sensitivity to Wnt3a. We have added this new data in Figure 5d and modified the text accordingly.

13. *Suppl.Fig.1b It would be worthwhile showing a better quality image to recognize peaks more clearly.*

RESPONSE

This figure has been adjusted so that the peaks are clearly visible.

14. Suppl.Fig. 6a It would be very helpful to indicate the origin / type of cell line that was tested (as an additional column within the table).

RESPONSE

We amended the table in Figure 6a to include a brief description of the cell lines.

15. Text line 315 hiPSCs are mentioned to have been tested. However, they are not indicated in any figure. If the authors have data based on hiPSCs it would be worthwhile to show them. The data would increase the impact of the manuscript beyond hPSCs.

RESPONSE

We have used hiPSCs in various studies and have found no significant differences compared to hESCs. For brevity sake, we are not including these data in this study, since they will not alter our overall conclusions, and we have deleted the reference to hiPSCs in the Discussion. It should be noted that the majority of studies have concluded that hiPSCs and hESCs are virtually indistinguishable from another.

16. The title may not really reflect what the paper shows; would it not be more accurate to use a variation of the running title?

RESPONSE

We agree that the use of the term “silences” in the title may be misleading. We therefore decided to change the title to “The WNT target SP5 negatively regulates expression of WNT transcriptional programs”.

17. The abbreviation for the hPSCs should appear in text line 4 for the first time and not in line 38.

RESPONSE

This change has been made.

18. Text line 51: The duration and method (recombinant protein, plasmid?) of the treatment should be mentioned.

RESPONSE

This information has been added in the revised manuscript.

Reviewers' comments:

Reviewer #1 (Remarks to the Author):

my concern is not satisfactorily addressed.

First, Supplemental Figure 2i and 2j remain the same in the revised manuscript. Authors should compare between WT GFP+ CXCR+ cells of and Sh or OE SP5 GFP+ CXCR4+ cells; Also, it is not clearly stated whether experiments were conducted with cells from EB at day 10 or not. The number of experiments and SD or SEM for data analysis are not provided. If the cells in Supplemental Figure 2i and 2j were collected from day 10 EBs, I am surprised to see the subtle changes between control and genetically manipulated cells, as the differences in the expression levels of FOA2, MIXL1 and PAX6 between WT and Mutant cells are so drastic, as shown in Figure 2d.

Second, Figure 2 legend states SEM for 4 technical replicates. This reviewer believe that SD, instead of SEM, should be used. In addition, for significance analysis, data from at least three independent experiments should be used. The technical replicates are not independent experiments, unless that the authors did not state correctly.

Third, to characterize the differentiation state of a population of cells, it is not sufficient to only show the mRNA level of a few genes. A set of marker genes for each lineage should be tested dynamically to define the role of a gene for differentiation. More convincingly, protein levels or staining of marker genes should be shown.

Reviewer #2 (Remarks to the Author):

I am pleased with the response of the authors and new data addressing my concerns. In the present form the manuscript represents a significant contribution to the field. I therefore recommend to accept the manuscript for publication in Nature Communications.

REVISION 2 REBUTTAL TO REVIEWER COMMENTS

Reviewer #1 (Remarks to the Author):

My concern is not satisfactorily addressed.

RESPONSE

The overarching theme of this reviewer's critique relates to the characterization of the differentiation potential of hPSCs in which SP5 expression has been altered, either by gene knockout, knockdown or overexpression. We provide a series of complementary experiments, including new data in this submission, that all support our conclusion that SP5 influences the transcriptional landscape of differentiating hPSCs:

1. Using embryoid body formation, SP5 mutant cells (dZF clones 1 and 2), exhibit a distinct differentiation profile compared to their WT counterparts, as evidenced by morphological analysis (Figure 2c, Supplementary Figure 2f, g) and gene expression analysis (Figure 2d, Supplementary Figure 2h).
2. **NEW DATA:** Using a monolayer differentiation protocol of WT and dZF cells followed by whole transcriptome analysis by RNA-Seq, we find that WT and dZF transcriptomes increasingly diverge over the course of differentiation (3 days versus 10 days). We illustrate this divergence in differentiation in two correlation matrices that are included as Figure 2e. This analysis demonstrates that the expression of over 5,000 genes diverges significantly in SP5 mutant cells relative to WT cells. The two mutant clones exhibit tight correlation (0.97 at Day 3 and 0.90 at Day 10) with each other. In contrast, each mutant line diverges significantly from WT cells (0.83-0.88 at Day 3 and 0.42-0.61 at Day 10).
3. Directed endoderm differentiation.
 - A. Using a well-established protocol to promote endoderm differentiation (depicted in Figure 2f), we show that expression of several endodermal markers is significantly altered in SP5 mutant cells relative to WT cells (Figure 2g).
 - B. We use shRNA-mediated knockdown (KD) and SP5 overexpression (OE) to demonstrate that SP5 levels influence the percentage of cells with endodermal characteristics, as monitored by flow cytometry for CXCR4 (Updated Supplementary Figure 2i and j). A challenge in this experiment was that SP5 KD or OE was never uniform across the cell population. This phenomenon of variable transgene expression has been observed by many other groups and is attributed to transgene silencing in human pluripotent stem cells (for examples see PMIDs: 17348812, 17379764, 27795446). In our particular experiments, we use GFP expression as a surrogate marker to detect cells expressing highest levels of either the shRNA or the SP5 cDNA. By comparing GFP positive and GFP negative cells, we are able to observe a clear effect of both SP5 KD and OE: cells with SP5 knockdown express higher levels of the endodermal marker CXCR4, whereas cells with SP5 overexpression express lower amounts of CXCR4. Therefore, SP5 acts to limit endodermal differentiation in this setting.

We contend that these independent lines of investigation provide compelling evidence that SP5 regulates the transcriptomes of differentiating hPSCs.

First, Supplemental Figure 2i and 2j remain the same in the revised manuscript. Authors should compare between WT GFP+ CXCR+ cells of and Sh or OE SP5 GFP+ CXCR4+ cells.

RESPONSE

We addressed this point above. We would like to add here that we have repeated this experiment (both KD and OE) a total of 4 times, each time obtaining the same result. Therefore, this result is highly reproducible. Furthermore, it is not independent from, but complements our results obtained with WT and SP5 mutant cells throughout this manuscript.

Also, it is not clearly stated whether experiments were conducted with cells from EB at day 10 or not.

RESPONSE

The experiment in question here is directed differentiation into endoderm, not EB formation. To clarify this in the figure we added short descriptive titles to the panels of Figure 2: “**d Embryoid body differentiation**”; “**e Monolayer differentiation**”; “**f Endoderm differentiation**”.

The number of experiments and SD or SEM for data analysis are not provided. If the cells in Supplemental Figure 2i and 2j were collected from day 10 EBs, I am surprised to see the subtle changes between control and genetically manipulated cells, as the differences in the expression levels of FOA2, MIXL1 and PAX6 between WT and Mutant cells are so drastic, as shown in Figure 2d.

RESPONSE

Supplementary Figures 2i and j are 4-day directed endoderm differentiation experiments, not EB formation experiments. Although all of these differentiation experiments point to the same conclusion, namely that SP5 regulates differentiation of hPSCs, the results from each method of differentiation (EB formation, monolayer differentiation, and directed endoderm formation) cannot be directly compared.

Second, Figure 2 legend states SEM for 4 technical replicates. This reviewer believe that SD, instead of SEM, should be used. In addition, for significance analysis, data from at least three independent experiments should be used. The technical replicates are not independent experiments, unless that the authors did not state correctly.

RESPONSE

We respectfully disagree with the reviewer on the need for SD of biological replicates for RT-qPCR data. First, the choice of metric for the representation of error bars does not affect the result of the t-test for significance. Second, the presentation of RT-qPCR data as the mean of technical replicates with error represented by SEM has precedence in the literature. Some recent examples in publications in Nature Communications where RT-qPCR data was represented as a mean of as few as 3 technical replicates with SEM error bars include:

Edri et al, Nat Commun. 2015 Mar 23;6:6500, PMID: 25799239

Abe et al, Nat Commun. 2015 May 7;6:7052, PMID: 25948511

van der Heijden M et al, Nat Commun. 2016 Mar 9;7:10916, PMID: 26956214

Third, to characterize the differentiation state of a population of cells, it is not sufficient to only show the mRNA level of a few genes. A set of marker genes for each lineage

should be tested dynamically to define the role of a gene for differentiation. More convincingly, protein levels or staining of marker genes should be shown.

RESPONSE

We agree with the reviewer that using a small number of genes to characterize the differentiation state of WT and SP5 mutant (dZF1 and 2) lines is insufficient. To address this point comprehensively, we used RNA-seq to compare the transcriptomes of WT and dZF cells that had been differentiated in a monolayer for 3 and 10 days. This data is included in the form of two correlation matrices in Figure 2e. This data provides confirmation that WT and SP5 mutant cells exhibit significant differences in their differentiation behavior.

In regards to protein expression, we provide flow cytometry of CXCR4 expression as a read-out of endoderm differentiation (Supplementary Figure 2i and j). Because this work fundamentally aims to describe transcriptional regulatory circuitry, we contend that the wealth of gene expression analysis, both by RT-qPCR (6 markers, 2 for each germ layer, Figure 2d and Supplementary Figure 2h) and RNA-seq (Figure 4a and new data summarized in correlation matrices in Figure 2e), provide sufficient characterization of the differentiation state of WT and SP5 mutant cells, and that further, while interesting from a developmental biology point of view, exceeds the scope of the work and potentially the page, word and figure limitations for this journal.

Reviewer #2 (Remarks to the Author):

I am pleased with the response of the authors and new data addressing my concerns. In the present form the manuscript represents a significant contribution to the field. I therefore recommend to accept the manuscript for publication in Nature Communications.

RESPONSE

Thank you for the comprehensive and helpful review.

REVISION 1 REBUTTAL TO REVIEWER COMMENTS

Reviewer comments are italicized, responses are not.

Reviewer #1 (Remarks to the Author):

1. The novelty of finding is not strong. It has been already reported that SP5 is a downstream target of Wnt signaling, acting as either a transcriptional repressor or activator. Moreover, the relationship between SP1 and SP5 downstream of Wnt signaling described here is also known. In particular, a recently published study has reported SP5 DNA binding profiles in mouse embryos and differentiating ESCs.

RESPONSE

Our study focuses on a system in which SP5 is found to be exclusively a repressor. Further, it is found here for the first time to be highly selective in where it binds DNA to impose that repression. The prior work noted by this reviewer laid the groundwork for understanding SP5's importance but left open major ambiguities, including whether it is a repressor or activator or both, under what conditions, and of what genes, and what is the critical outcome of the repression. The results reported here eliminate the prior ambiguities and underscore the critical importance of SP5 as a negative regulator of the Wnt signaling pathway at the level of its targets, rather than at the level of pathway components. Thus, although the reviewer raises several important and valid points of criticism, we respectfully disagree with the assertion that the novelty of these findings is not strong.

To our knowledge, our comprehensive study is the first of its kind to perform and analyze a time series of signaling in human pluripotent stem cells. We integrate genome-wide transcriptional and chromatin data to identify a previously unknown mechanism by which Wnt, a key developmental growth factor, regulates developmental programs. The model that emerges from our studies, namely that SP5 acts to negatively regulate expression of the entire Wnt transcriptional program, is a conceptual advance that is of significant interest and importance to the community of developmental and stem cell biologists. The concept of developmental competence has been the subject of many years of intense research in developmental biology. Our study contributes to this field by demonstrating that mechanisms exist that dampen the transcriptional response to developmental regulators, in this case Wnt. Such dampening of transcriptional programs is arguably equally as important in differentiation as the initial establishment of developmental competence. The reviewer is correct that other studies have examined the role of SP5 in Wnt signaling. Our study builds on these prior studies and puts forth a model that has not been proposed previously. We further contend that our model will have implications beyond Wnt signaling, as other developmental signaling pathways may be regulated by a similar mechanism.

2. The evidence that SP5 restricts endoderm formation is not solid. Data presented in figure 2f, 2g and Supplemental Fig 2i are not convincing to make conclusion that SP5 acts to regulate the endoderm formation. It seems that there is a problem in analyzing data of Fig 2g and Supplemental Fig 2i.

RESPONSE

What is indisputable is that SP5 regulates differentiation, and the findings here report this for the first time. This comment from the reviewer helped us realize that Fig. 2g and Supplementary Fig. 2i (Supplementary Fig. 2i and j in the revised version) were laid out

in a complicated manner that made the data difficult to follow. The effects on endodermal differentiation are quite subtle, and that contributed to the difficulty in reading/following the figure. To clarify the paper and highlight the most important point, that SP5 indeed regulates differentiation, we have focused our conclusions and interpretations of the Figure 2 experiments on this point.

Our data demonstrates solidly that SP5 regulates differentiation. To draw this conclusion, we now present several independent pieces of data: (1) the transcriptional response of SP5 mutant cells is distinct from that of WT cells; (2) SP5 mutant cells exhibit a differentiation defect compared to WT cells; (3) SP5 mutant cells show defects even during directed differentiation; (4) shRNA-mediated knockdown of SP5 also shows differentiation defect; and (5) SP5 overexpression diminishes differentiation, shown here for the case of endoderm. Together, we respectfully submit that our central conclusion that SP5 regulates hPSC differentiation, is warranted and well-supported.

With respect to analysis of data in Fig. 2g and Supplementary Fig. 2i (now consolidated into Supplementary Fig. 2i and j), we direct this reviewer to our response to comment 8 by reviewer 3.

3. Results in Fig 5 are most descriptive, lacking molecular insights and functional consequences.

RESPONSE

The purpose of Figure 5 is to delineate the stem cell context of SP5's role as a post-hoc negative regulator of Wnt signaling – post-hoc in the sense that it dampens the up-regulated targets of Wnt signaling in a time-sensitive manner. The reviewer is correct in noting the results in Figure 5 are descriptive, but that is the goal: to characterize the stem cell context in which SP5 has an essential role. Although we do not prove that SP5 is a stem cell-specific Wnt target gene—and we make no such bold claim—we provide substantial evidence that induction of SP5 in response to Wnt signaling correlates with stemness properties. This observation will be of interest to the scientific community and help guide others as they study SP5 in their own systems. The publication of the correlation between SP5 induction and stemness as part of this study is of value. Several other studies and reviews have suggested that SP5, like AXIN2, is a universal Wnt target gene. Our studies challenge this claim and indicate that SP5 expression in response to Wnt signaling is more selective than that of AXIN2. These findings, despite their descriptive nature, will be of significant interest to researchers studying developmental processes that are affected by Wnt signaling. For example, while Axin2 is commonly used as an in vivo reporter for Wnt signaling and for lineage tracing experiments in mice, similar experiments with Sp5 will likely yield significant differences. Therefore, our observation that SP5 expression in response to Wnt signaling is highly cell context dependent represents an important and valuable contribution.

Reviewer #2 (Remarks to the Author):

QUESTION/COMMENT

1. First of all, the mechanism for restricted targeting of SP5 (662 SP5 binding sites compared to 24,747 SP1 sites) was not investigated. How Sp5 is recruited to only a limited set of genomic targets significant number of which are Wnt/b-catenin signaling target genes? Given the consensus/logo being shared by SP5 and SP1 the targeting process has to be highly regulated. Kennedy et al. show that Sp5 interacts with Tcf/Lef.

Does Sp5 bind to Tcf/Lef in Wnt3a-stimulated hPSC to be recruited near Wnt-target genes? Authors have anti-SP5 antibody, so they might be able to perform time-course co-immunoprecipitations in Wnt3a-stimulated hPSC.

RESPONSE

This is indeed a critical point that we had not addressed in the original version of the paper. Here we provide additional data that sheds light on the likely mechanism by which SP5 binding is highly selective compared to SP1, a ubiquitously acting transcription factor. The observation that SP5 targets WNT target genes and subsequently represses their expression suggested that SP5 interacts with core components that mediate the transcriptional response to a WNT signal, specifically beta-catenin and/or LEF/TCF. To test this possibility, we used a proximity ligation method called APEX2 (Lam et al., Nat Methods 12: 51-4, 2015, PMID: 25419960), which uses reactive biotin to label proteins within a 10nm radius of the APEX2-tagged protein. Using this method we find that SP5-APEX2 leads to labeling of beta-catenin, providing strong evidence that these two proteins are in close proximity. As a control for specificity of this highly specific labeling method, we show that the abundant protein beta-actin is not labeled. We have incorporated this data in the revised manuscript in Supplementary Figure 4c.

Of note, we attempted to show this interaction by co-immunoprecipitation of endogenous SP5 and beta-catenin, however, we were unable to do so. We attribute this to the fact that SP5 is a nuclear protein and conditions required to detect endogenous SP5 require harsh lysis conditions that disrupt most protein-protein interactions. These harsh lysis conditions are not problematic for the ChIP-seq experiments described in our paper since cells are cross-linked with paraformaldehyde, which stabilizes the SP5-DNA complexes.

In addition, it should also be noted that a previous publication provided evidence for Sp5 and beta-catenin binding (Kennedy et al, PNAS, PMID: 26969725), further substantiating our claim for this interaction.

2. SP5 promoter is a target of SP5 protein upon Wnt3A pathway stimulation at 24h. Given the mechanism proposed here as a main outcome of the paper it follows that SP5 repressor should start to diminish SP5 gene expression. Yet, at 48 hours transcriptomics data show an increase of SP5 transcription in WT cells (Fig1f). Authors should extend the time window of transcriptional profiling in order to see downregulation of SP5 gene later on during continuous (or pulse) Wnt3A stimulation. Isn't this the way to show the termination phase for the model shown in Fig.6 by experimental means?

RESPONSE

This is an excellent point. We have performed a longer time-course of Wnt3a treatment and observe a robust down-regulation after 4 days of treatment. Furthermore, and very importantly, this down-regulation of SP5 is not observed in SP5 mutant cells. We have included this data in Figure 4f with the associated text in the manuscript. Together these data provide strong evidence that SP5 feeds back to repress expression of Wnt target genes, including itself.

Minor comments.

3. The core of the paper is the ChIP-seq using a novel proprietary SP5 antibody generated against N-terminus of SP5. This region contains a short stretch of homology (aa31-about aa50) to other zinc finger proteins (peptide SPLA....PG). It makes sense to provide a small characterization panel (western, IP) for this essential tool as a

supplementary information. Please state consistently if aa1-129 (Methods) or aa1-133 (Supplementary Fig. 1f) is correct. Also, instead of stating in Methods as a source of antibody (line 490) ...provided by Abcam... it is more appropriate to saycustom generated by Abcam.

RESPONSE

The SP5 antibody used in these studies has recently become available through Abcam (Anti-Sp5 antibody, ab209385). The exact identity of the immunogen used to generate this antibody was never revealed to us and the sequence remains proprietary, as indicated in the data sheet. The now-commercially-available monoclonal antibody was used for Western blotting and immuno-fluorescence (Figures 1d, 1e, 2b).

Prior to the isolation of this monoclonal antibody, we obtained small samples of crude serum from rabbits that had been immunized with the proprietary N-terminal synthetic peptide. Knowing that the antibody was raised to some portion of the N-terminus of SP5, we generated a GST fusion protein with amino acids 1-129 of SP5, and used this fusion protein to affinity purify the crude antiserum. The affinity-purified antiserum was then used for the ChIP-seq experiments. Given the extremely time consuming and costly process for ChIP-seq experiments, we respectfully submit that repeating these experiments with the new SP5 monoclonal antibody would be excessive. Most importantly, such repeat experiments will not alter the conclusions of our manuscript. We have corrected Supplementary Fig. 1f to indicate aa1-129.

4. The authors made a considerable effort on investigating the mechanism for an interesting observation that SP5 is significantly more Wnt-inducible in stem-like cells and during differentiation the SP5 ability to be induced by Wnt signalling diminishes. Authors conclude that changes in poised chromatin likely accounts for the loss of SP5 induction in response to Wnt3A. Why were H3K27me2 and H3K4me2 marks used for ChIP-PCR in Sp5 promoter region(Fig 5)? H3K4me1 and H3K4me2 are enriched at enhancers and H3K4me3 is enriched at promoters (reviewed in Chen and Dent), so maybe H3K4me3 should be checked. Since H3K27me3 surprisingly showed non-significant change during differentiation H3K9me3 that also marks facultative heterochromatin should be examined.

RESPONSE

The mechanism by which the SP5 promoter is silenced in differentiated cells relative to undifferentiated cells is certainly of interest, and we provide additional data as requested by this reviewer. Specifically, we show that the promoter mark H3K4me3 is highly enriched in the SP5 promoter in undifferentiated hES cells compared to Day 27 differentiated cells this data in Figure 5d. The data for the chromatin mark H3K27me2, previously in Figure 5d, has been moved to Supplementary Figure 6e.

5. Why is the TSS+500 window not the same scale for all genes in Supp.Fig1b?

RESPONSE

The reason for this was that this histone mark as visualized using these wiggle plots varied substantially from gene to gene. However, we agree with the reviewer that this makes it difficult to look at and we have adjusted the figure such that the scale is identical for all genes shown. The important point that this histone mark is enriched upon Wnt3a treatment at these loci is clearly visible.

6. Perhaps it makes more sense to have both loss- (shSP5-GFP) and gain-of function

(SP5-GFP) experiment shown together in the supplement or main Figure. In left panel of Fig2g the green and grey bar appear to be swapped based on numbers in the left panel.

RESPONSE

We agree that showing this data together will make it clearer. We therefore consolidated the loss- and gain-of-function experiments in Supplementary Figure 2i and j.

7. Indicate what red dots (more stem-like) and grey dots stand for in Fig5a.

RESPONSE

We have modified this figure to make the point more clearly, and we have added additional text to the corresponding figure legend.

8. Heat map shows that other Klf/SP genes (Klf7, Klf8 and even SP8) are slightly induced during Wnt3A stimulation albeit their absolute levels (RPKM) are much lower than that of SP5. What is the induction of the respective genes between 0h and 48 hours?

RESPONSE

The reviewer is correct that these members of the KLF/SP family are upregulated. However, the fold induction is significantly lower than that of SP5. The RPKMs from our RNA-seq experiment, which is provided in Supplementary Table 1, details levels of induction.

Gene ID	Hours Wnt3a				Fold induction (min vs max)
	0	12	24	48	
SP8	3.2	5.1	5.2	9.9	3.1
KLF7	9.5	11.5	16.3	17.3	1.8
KLF8	5.0	4.0	4.7	13.3	2.7

9. Line 118.....SP5 mutant cells expressed significantly higher levels of the DE markers SOX17 and CXCR4 compared to WT cells (Fig. 2f)..... well yes, but not at all stages, please re-phrase

RESPONSE

The text has been modified to indicate that expression of DE markers in SP5 mutant cells is elevated relative to WT cells.

Reviewer #3 (Remarks to the Author):

Criticism and question (not sorted according to their importance):

1. Fig.1a (and others): The authors used 1nM Wnt3a. How was this concentration calculated, i.e. the molecular weight determined of Wnt3a? Functional Wnt protein is heavily post-translationally modified (with possibly complex glycosylation pattern).

RESPONSE

We generate our recombinant Wnt proteins in house using methods that we have developed and optimized (e.g. see PMID: 19099243), and determine the concentration of each batch. To get an approximate value for the concentration of a recombinant Wnt3a protein batch, we visualize the purified protein on a Coomassie-stained gel. Using a dilution series of a protein standard (e.g. BSA from 10 ng to 1ug), we are able to estimate the concentration of Wnt3a in ug/ml. Wnt3a migrates as single band—albeit somewhat fuzzy band which is likely due to variable amounts of glycosylation—at ~40kDa. Using these values for MW and concentration, we convert ug/ml to molarity. In general, our Wnt3a stock solution is at a concentration of 1-5 uM (40ug/ml – 200ug/ml).

2. Fig.1a (and other Figs.) The changes in morphology may not necessarily reflect the pluripotent/differentiated status of stem cells (reported for example in b-catenin-null mESCs that look as differentiated, but still express key stem cell markers). Although hPSCs are of different origin (and stage) it would be worthwhile to show immunofluorescent staining (stem cells vs. differentiation markers and/or some targets/GOIs). Immunostaining would also increase the impact of other figures (Fig. 2d,e,f and Fig.4 a,b), since changes in mRNA level may not fully reflect protein levels.

RESPONSE

We agree with the reviewer that the morphological changes in response to Wnt3a treatment are subtle. We therefore included additional data to show by immunofluorescence and flow cytometry both the acquisition of differentiation markers (SOX17 in Figure 1b) and the loss of pluripotency markers (FZD7, SSEA4 and TRA1-81 in Supplementary Figure 1a and b).

3. Fig.1b and Fig.4a would benefit by indicating directly within the figure(s) or at least in the figure legend what particular classes/groups of genes (i – iv) are represented; this would facilitate the reading.

RESPONSE

We have added these descriptions to the figures (Figure 1c and 4a) to clarify the individual gene sets.

4. Fig1e Can the authors show a higher magnification (i.e. to show the detail/inset in higher magnification)? In some cells it is hard to see if the SP5 is really nuclear.

RESPONSE

We have included higher resolution images and a higher magnification of the inset to clearly show that SP5 is primarily localized to the nucleus.

5. Fig2b Could be moved (especially the lower panel) to a supplementary figure.

RESPONSE

Thank you for this suggestion, however, we feel that showing protein levels and the longer exposure of the same immunoblot is important, as it demonstrates that very little truncated SP5 protein is present in the mutant cells. Such truncated proteins, if abundant, could exert dominant effects. Given the low amounts of these truncated proteins, which is clearly documented in this figure, we contend that the SP5 mutation is a complete loss of function mutation.

6. Fig.2e,f If mutating SP5 pushes hPSCs cells into endodermal (DE) differentiation, are

the other branches of differentiation affected in dZF mutants (e.g. differentiation into neuroectoderm etc.)? Did author check this?

RESPONSE

The main point of the data in this figure is to show that differentiation of SP5 mutant hESCs is impaired compared to wildtype hESCs. Consistent with our model we observe a significant shift towards mesendodermal differentiation. In this revised version we provide additional data (Figure 2d) that this mesendodermal skewing is accompanied with a concomitant loss of ectodermal differentiation, as highlighted by a significant elevation of PAX6 expression in WT cells compared to SP5-mutant cells.

7. Fig.2f (and main text lines 118 – 121). It is mention within the text:...” SP5 mutant cells expressed significantly higher levels of the DE markers SOX17 and CXCR4 compared to WT cells (Fig. 2f). In contrast, expression of the mesendodermal marker T was similar in WT and dZF cells...”.

Although the differences in expression the data seems to be significant, due to very consistent levels as apparent based on the low error bars showing SEM or SD (please indicate), the significance reduces over time and the variations are, in general, not so dramatic. For instance, CXCR4 in dZF1 cells is expressed at the same level as in wt cells after 3 days (end of differentiation protocol). On the other hand, T is expressed significantly lower at 2 days. This is the time point, when also the differences in expression of other targets are most prominent. Can the authors comment on this? Or change the text to be more precise?

RESPONSE

We agree with the reviewer that the effects of SP5 knockout (KO) on endodermal differentiation are subtle, especially at later time points. In the revised manuscript, we address this point, emphasizing that SP5 KO significantly alters the differentiation potential of hPSCs. While there is a clear skewing towards mesendodermal lineages during non-directed differentiation and towards DE during directed differentiation, the primary effect we wish to highlight is the difference in differentiation potential between WT and SP5 mutant cells.

Error bars are SEM for 4 technical replicates.

8. Fig.2g The more than 10% difference in GFP(-) cells between shControl and shSP5 is close to that one observed in the middle panel (14,9%). Should the percentage of GFP(-) cells expressing gated CXCR4 level not be more comparable between experiments? If it is so variable can the authors perform the following experiment: to use different labeling for shControl and shSP5 (e.g. red and green) mix the cells together and compare really shControl and shSP5 within one experiment?

RESPONSE

The efficiency of hPSC differentiation into DE is highly variable and dependent on various factors, such as seeding density and cell line. The slight differences in DE formation between the various differentiations, ranging from 40.1 to 64.1% (Supplementary Figure 2i and j), do not influence the interpretation of the data. For each experiment—shRNA-mediated knockdown (KD) and overexpression (OE)—the important difference is between the GFP-positive population (= transduced cells) relative to the GFP-negative population (= non-transduced cells). In the case of KD (Supplementary Figure 2i), CXCR4 expression is elevated in the GFP-positive population of SP5-shRNA transduced cells, an effect not observed for the control shRNA

transduced cells. Conversely, OE of SP5 produces the opposite effect, i.e. a decline in CXCR4 expressing cells in SP5 overexpressing population (Supplementary Figure 2j). Given the reproducibility of this experiment (we have performed this experiment 3 times with the same result), the complementary results for KD and OE, and the similar effects observed for dZF cells, we feel that these experiments convincingly demonstrate that SP5 influences the differentiation potential of hPSCs.

9. Fig.4a The class/group (iv) of the Wnt-target genes is most strongly upregulated after 48 hours in both wt and dZF cells (however in dZF1 it is higher). If SP5 accumulates over time and starts to repress the transcription of target genes should the level of transcripts not be lower in wt (and not highest) at the last time point (differentiated cells)? Or it is just a matter of duration and later it will decrease in wt cells (and not in mutants)? Can authors comment on this?

RESPONSE

We appreciate this astute observation. Gene expression in response to Wnt3a is highly dynamic, in part because of the negative feedback exerted by SP5, which we describe in this manuscript. In the SP5 mutant cells, the normal brakes that restrain expression of Wnt target genes are released and expression of these genes increases compared to WT cells. Furthermore, it follows, as pointed out by the reviewer, that in WT cells expression of these Wnt targets should decline as cells proceed through differentiation. To address this point, we performed a longer time course of Wnt3a treatment and found that expression of SP5 declines after 3 days, consistent with negative feedback regulation. Furthermore, this decline is not observed in SP5 mutant cells, further supporting the idea that the normal brakes are lacking. We have included this new data in Figure 4f.

10a. Fig. 4b, e If dZF mutants express SP5 that cannot bind genomic DNA (did the authors test this?)

RESPONSE

We have not tested whether the mutant SP5 protein binds genomic DNA, however, given that the entire zinc finger domain is deleted and very little mutant protein is produced, we find it highly unlikely that this truncated protein has any DNA binding activity.

10b. Fig. 4b, e If dZF mutants express SP5 that cannot bind genomic DNA (did the authors test this?) what happens at the regions of SP5-dependent genes, at loci where SP5 peaks are present in wt cells? If SP5 can not bind there (and thus outcompete SP1) the regulatory site may remain free for SP1. Is SP1 bound there (i.e. as in wt unstimulated cells). It would be worthwhile to show SP1/SP5 ChIP for some targets (e.g. T, SP5, TBX3...). At least the authors should comment on this. (TCF or b-catenin ChIP might be also very interesting to show).

RESPONSE

This is in fact an interesting experiment, one that we have attempted. We performed SP1 ChIP from WT and dZF cells followed by qPCR for the promoter regions of *SP5* and *T*. Results are unfortunately inconsistent and difficult to explain. A representative experiment is shown below. The increased SP1 occupancy at the *SP5* promoter in dZF cells is potentially interesting, however, we do not observe this effect for the *T* promoter. One potential explanation for the effect at the *SP5* promoter may be that CRISPR/Cas9-

mediated deletion of a large portion of the *SP5* gene may have altered the local chromatin structure at this locus. Overall, these experiments are inconclusive and fail to provide additional insights into the competition between SP1 and SP5. In light of our many other results (including Figure 3f and Supplementary Figure 3a), as well as those of others (e.g. Fujimura et al. J Biol Chem 282: 1225-37, 2007), we respectfully submit that the conclusion of SP1-SP5 competition at select sites has strong support.

11. Fig.5 c The authors claim that the ratio of AXIN2 in *Wnt3a*-treated vs. unstimulated cells changed only slightly (main text line 227-228). In fact, 28 days after treatment the expression of AXIN2 is insensitive to *Wnt3a* stimulation comparable to SP5. What is different is the dynamics of the process (“sensitivity” of SP5 decreases during time, AXIN2 goes up and decreases afterwards). Can the authors comment on this (or change the text accordingly)?

RESPONSE

The reviewer is correct in stating that expression of AXIN2 in response to *Wnt3a* is comparable to that observed for SP5 in differentiated cells. The main point of this experiment is to show the desensitization in SP5 expression in response to *Wnt3a*; for these experiments, AXIN2 served as a “control” since it is considered to be a universal *Wnt* target gene. In contrast to SP5, AXIN2’s inducibility is only marginally affected. We have rephrased the text such that it more accurately describes the AXIN2 data.

12. Can the authors speculate more (or, better, can they check) about the nature of this *Wnt*-desensitisation? For example: are some secreted inhibitors expressed (sFRPs, Dkk), what is the nature of TCF/LEFs (in mESCs a TCF/LEF switch was described), is Tankyrase more active.....

RESPONSE

Our data indicate that SP5 induction in response to *Wnt* is silenced through loss of epigenetic marks that are associated with active promoters, specifically H3K4me2 and H3K4me3, as shown in Figure 5e. However, other mechanisms, such as those suggested by the reviewer may be at play here as well. We performed the following experiment to further address this possibility: rather than stimulate the pathway with *Wnt3a*, we used a GSK3 inhibitor, CHIR98014, which acts downstream of *Wnts* and their receptors. Like *Wnt3a*, CHIR failed to potently activate *SP5* transcription in 28-day differentiated cells, indicating that this de-sensitization is not due to reduced sensitivity to *Wnt3a*. We have added this new data in Figure 5d and modified the text accordingly.

13. Suppl.Fig.1b It would be worthwhile showing a better quality image to recognize peaks more clearly.

RESPONSE

This figure has been adjusted so that the peaks are clearly visible.

14. Suppl.Fig. 6a It would be very helpful to indicate the origin / type of cell line that was tested (as an additional column within the table).

RESPONSE

We amended the table in Figure 6a to include a brief description of the cell lines.

15. Text line 315 hiPSCs are mentioned to have been tested. However, they are not indicated in any figure. If the authors have data based on hiPSCs it would be worthwhile to show them. The data would increase the impact of the manuscript beyond hPSCs.

RESPONSE

We have used hiPSCs in various studies and have found no significant differences compared to hESCs. For brevity sake, we are not including these data in this study, since they will not alter our overall conclusions, and we have deleted the reference to hiPSCs in the Discussion. It should be noted that the majority of studies have concluded that hiPSCs and hESCs are virtually indistinguishable from another.

16. The title may not really reflect what the paper shows; would it not be more accurate to use a variation of the running title?

RESPONSE

We agree that the use of the term “silences” in the title may be misleading. We therefore decided to change the title to “The WNT target SP5 negatively regulates expression of WNT transcriptional programs”.

17. The abbreviation for the hPSCs should appear in text line 4 for the first time and not in line 38.

RESPONSE

This change has been made.

18. Text line 51: The duration and method (recombinant protein, plasmid?) of the treatment should be mentioned.

RESPONSE

This information has been added in the revised manuscript.

Reviewers' comments:

Reviewer #1 (Remarks to the Author):

Authors have addressed the most of my concerns. However, one statistical problem remains. As I pointed out in my initial review comment, statistical comparison of marker gene expression levels between WT and mutant cells should be made using data collected from at least three independent experiments, i.e., three biological replicates, instead of technical replicates, no matter how many technical replicates in single experiments and whether SD or SEM are used. However, authors in this study (Fig. 2g) compared the mRNA levels of marker genes between WT and mutant cells with data collected from 4 technical replicates in one single experiments. The misuse of the analysis makes very small differences between samples within single experiments statistically significant and the variance among samples is usually small. For example, the difference in SOX17 mRNA levels is obviously very small between WT and mutant cells at day 3 in Fig. 2g. However, their analysis result is $p < 0.001$. People would question such analysis results.

Generally, the variance from independent experiments of differentiating human pluripotent stem cells is large, which is why a statistical analysis is needed to determine whether the difference between different samples are statistically significant or not. However, the difference among replicates of single experiments (technical replicates) is usually very small, as seen here, and could not be used to evaluate whether the result is biologically representative or whether the result is only true for a particular single experiment.

In sum, I have clearly stated my opinion. Authors argue as there is the precedence in the literature. However, the precedence is not necessarily correct. It is dependent on whether the journal cares its reputation or not.

REVISION 3 REBUTTAL TO REVIEWER COMMENTS

Reviewer #1 (Remarks to the Author):

Authors have addressed the most of my concerns. However, one statistical problem remains. As I pointed out in my initial review comment, statistical comparison of marker gene expression levels between WT and mutant cells should be made using data collected from at least three independent experiments, i.e., three biological replicates, instead of technical replicates, no matter how many technical replicates in single experiments and whether SD or SEM are used. However, authors in this study (Fig. 2g) compared the mRNA levels of marker genes between WT and mutant cells with data collected from 4 technical replicates in one single experiments. The misuse of the analysis makes very small differences between samples within single experiments statistically significant and the variance among samples is usually small. For example, the difference in SOX17 mRNA levels is obviously very small between WT and mutant cells at day 3 in Fig. 2g. However, their analysis result is $p < 0.001$.

People would question such analysis results.

Generally, the variance from independent experiments of differentiating human pluripotent stem cells is large, which is why a statistical analysis is needed to determine whether the difference between different samples are statistically significant or not. However, the difference among replicates of single experiments (technical replicates) is usually very small, as seen here, and could not be used to evaluate whether the result is biologically representative or whether the result is only true for a particular single experiment.

In sum, I have clearly stated my opinion. Authors argue as there is the precedence in the literature. However, the precedence is not necessarily correct. It is dependent on whether the journal cares its reputation or not.

RESPONSE

Although it is possible for us to complete the necessary experiments to address this important point of criticism, such experiments represent a significant burden, both in terms of time and resources required. More importantly, successful completion of these follow-up experiments will in no way alter the conclusions of this paper. We therefore propose deleting the data in question from this manuscript.

The main point is that SP5 mutant cells exhibit defects in their ability to differentiate. Aside from the endodermal differentiation experiments, which we propose to delete from this manuscript, we provide four independent lines of experimentation that support our conclusion that SP5 influences the differentiation potential of hESCs:

- (i) Differentiation of SP5-deficient cells yields abnormal embryoid bodies (EBs) (Figure 2c, Supplementary Figure 2e-g).
- (ii) EBs derived from wildtype and SP5 mutant cells exhibit significant differences in gene expression (Figure 2d, Supplementary Figure 2h).

- (iii) Monolayer differentiated WT and SP5 mutant cells exhibit significant differences in their transcriptome (Figure 2e).
- (iv) Expression of Wnt target genes that are targeted by SP5 (as shown by CHIP-Seq) is significantly increased in SP5 mutant cells (Figure 4a, e and f, Supplementary Figure 5).

Together, these data establish that SP5 mutant cells are impaired in their response to differentiation cues. Showing that endodermal differentiation is also perturbed does not alter this central conclusion.

In light of these data, we would like to remove the experiments on endodermal differentiation, specifically the data shown in Figure 2g and in Supplementary Figure 2i and j. Although we remain convinced of our findings that SP5 deletion, knockdown or overexpression perturbs endodermal differentiation, removing this data from this manuscript does not alter or weaken our conclusions.